# LAST ITERATE CONVERGENCE IN MONOTONE MEAN FIELD GAMES

## ABSTRACT

Mean Field Game (MFG) is a framework utilized to model and approximate the behavior of a large number of agents, and the computation of equilibria in MFG has been a subject of interest. Despite the proposal of methods to approximate the equilibria, algorithms where the sequence of updated policy converges to equilibrium, specifically those exhibiting last-iterate convergence, have been limited. We propose the use of a simple, proximal-point-type algorithm to compute equilibiria for MFGs. Subsequently, we provide the first last-iterate convergence guarantee under the Lasry–Lions-type monotonicity condition. We further employ the Mirror Descent algorithm for the regularized MFG to efficiently approximate the update rules of the proximal point method for MFGs. We demonstrate that the algorithm can approximate with an accuracy of $\varepsilon$ after $\mathcal{O}(\log(1/\varepsilon))$ iterations. This research offers a tractable approach for large-scale and large-population games.

## 1 INTRODUCTION

Mean Field Games (MFGs) provide a simple and powerful framework for approximating the behavior of large populations of interacting agents. Originally formulated by Lasry & Lions (2007); Huang et al. (2006), MFGs model the collective behavior of homogeneous agents in continuous time and state settings using Partial Differential Equations (PDEs) (Cardaliaguet & Hadikhanloo, 2017; Lavigne & Pfeiffer, 2023; Inoue et al., 2023). Subsequently, the formulation of MFGs using Markov Decision Processes (Bertsekas & Shreve, 1978; Puterman, 1994) has enabled the study of discrete-time and discrete-state models (Gomes et al., 2010), broadening the applicability of MFGs to Multi-Agent Reinforcement Learning (MARL) (Yang et al., 2018). Moreover, it has become possible to capture interactions among heterogeneous agents (Gao & Caines, 2017; Caines & Huang, 2019).

The applicability of MFGs to MARL drives research into their computational aspects. Under fairly general assumptions, the problem of finding an equilibrium in MFGs is known to be PPAD-complete (Yardim et al., 2024). Consequently, it would be essential to impose assumptions that allow for the existence of algorithms capable of efficiently computing an equilibrium. One of the assumptions is contractivity (Xie et al., 2021; Anahtarci et al., 2023; Yardim et al., 2023). However, it is known that many problems are not contractive in practice (Cui & Koeppl, 2021). One of the more realistic assumptions is monotonicity (Pérolat et al., 2022; Zhang et al., 2023; Yardim & He, 2024), which intuitively implies that as more

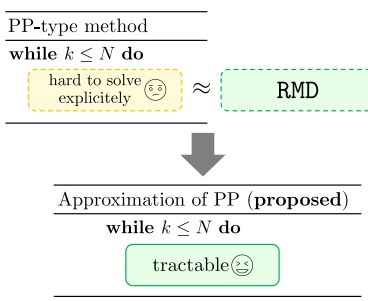

Figure 1: Overview of Algorithms

agents converge to a single state, the reward monotonically decreases. Under the monotonicity assumption, Online Mirror Descent (OMD) has been proposed and widely adopted (Pérolat et al., 2022; Cui & Koeppl, 2022; Lauriere et al., 2022; Fabian et al., 2023). OMD, especially when combined with function approximation via deep learning, has enabled the application of MFGs to MARL (Yang & Wang, 2021; Zhang et al., 2021; Cui et al., 2022).

Theoretically, *last-iterate convergence (LIC)*, which ensures that the policy obtained in the final iteration converges, is particularly important in deep learning settings due to the constraints imposed by neural networks (NN). In NNs, calculating the time-averaged policy like in the celebrated Fictitious

Play method (Brown, 1951; Perrin et al., 2020) may be less meaningful due to nonlinearity in the parameter space. These considerations have spurred significant research into developing algorithms that achieve LIC in finite $N$-player games, as seen in, e.g., Mertikopoulos et al. (2018); Piliouras et al. (2022); Abe et al. (2023; 2024).

Despite its importance, the literature on LIC results in MFG is quite limited. The only exception is Pérolat et al. (2022), who proved the LIC result for the continuous-time version of OMD without the quantified rates under the strict monotonicity condition. The aim of this research is to establish an online learning algorithm that can achieve LIC in MFGs under *non-strict* monotonicity conditions.

In this paper, we propose a novel proximal-point (PP) type algorithm and prove that it achieves LIC under the non-strict monotonicity assumption. Furthermore, we demonstrate that the update rule of the PP can be approximated efficiently by sequentially using the Regularized Mirror Descent (RMD). We further show that RMD achieves the approximation with the accuracy of $\varepsilon$ within $\mathcal{O}(\log(1/\varepsilon))$ iterations. Figure 1 summarizes the overview of the algorithms in this paper.

In summary, the contributions of this paper are as follows:

> **Contribution**
>
> (i) We construct the first algorithm based on the celebrated PP method that achieves LIC for general monotone MFGs (Theorem 4.3).
>
> (ii) We prove for the first time that regularized Mirror Descent achieves exponential convergence for monotone MFGs (Theorem 4.4).
>
> (iii) We combine these two algorithms as shown in Figure 1 to develop a tractable algorithm that approximates the PP-based method (Algorithm 2).

The organization of this paper is as follows: In Section 2, we review the fundamental concepts of MFGs. In Section 3, we introduce the PP method and its convergence results. In Section 4, we present the RMD algorithm and its convergence properties. Finally, in Section 5, we propose a combined approximation method, demonstrating its convergence through experimental validation.

## 2 SETTING AND PRELIMINARY FACT

### 2.1 NOTATION

For a positive integer $N \in \mathbb{N}$, $[N] := \{1, \ldots, N\}$. For a finite set $X$, $\Delta(X) := \{p \in \mathbb{R}_{\geq 0}^{|X|} \mid \sum_{x \in X} p(x) = 1\}$. For a function $f: X \to \mathbb{R}$ and a probability $\pi \in \Delta(X)$, $\langle f, \pi \rangle := \langle f(\bullet), \pi(\bullet) \rangle := \sum_{x \in X} f(x)\pi(x)$. For $p^0, p^1 \in \Delta(X)$, define the Kullback–Leibler (KL) divergence $D_{\mathrm{KL}}(p^0, p^1) := \sum_{x \in X} p^0(x) \log\left(p^0(x)/p^1(x)\right)$, and the total variation (TV) distance as $\left\| p^0 - p^1 \right\| := \sum_{x \in X} \left| p^0(x) - p^1(x) \right|$.

### 2.2 MEAN-FIELD GAMES

Consider a *Mean-Field Game (MFG)* that is defined through a tuple $(\mathcal{S}, \mathcal{A}, H, P, r, \mu_1)$. Here, $\mathcal{S}$ is a finite discrete space of states, $\mathcal{A}$ is a finite discrete space of actions, $H \in \mathbb{N}_{\geq 2}$ is a time horizon, and $P = (P_h)_{h=1}^H$ is a family of transition kernels $P_h: \mathcal{S} \times \mathcal{A} \to \Delta(\mathcal{S})$, that is, if a player with state $s_h \in \mathcal{S}$ takes action $a_h \in \mathcal{A}$ at time $h \in [H]$, the next state $s_{h+1} \in \mathcal{S}$ will transition according to $s_{h+1} \sim P_h(\cdot \mid s_h, a_h)$. In addition, $r = (r_h)_{h=1}^H$ is a family of reward functions $r_h: \mathcal{S} \times \mathcal{A} \times \Delta(\mathcal{S}) \to [0,1]$, and $\mu_1 \in \Delta(\mathcal{S})$ is an initial probability of state. Note that, in the context of theoretical analysis of the online learning method for MFG (Pérolat et al., 2022; Zhang et al., 2023), $P$ is assumed to be independent of the state distribution. It is reasonable to assume that at any time $h$, every state $s' \in \mathcal{S}$ is reachable:

**Assumption 2.1.** For each $(h, s') \in [H] \times \mathcal{S}$, there exists $(s, a) \in \mathcal{S} \times \mathcal{A}$ such that $P_h(s' \mid s, a) > 0$.

In this paper, we focus on rewards $r$ that satisfy the following two typical conditions, which are also assumed in Perrin et al. (2020; 2022); Pérolat et al. (2022); Fabian et al. (2023); Zhang et al. (2023). The first one is *monotonicity* of the type introduced by Lasry & Lions (2007), which means,

under a state distribution $\mu = (\mu_h)_{h=1}^H \in \Delta(\mathcal{S})^H$, if players choose a strategy—called a policy $\pi = (\pi_h)_{h=1}^H \in (\Delta(\mathcal{A})^{\mathcal{S}})^H$ to be planned—that concentrates on a state or action, they will receive a small reward.

**Assumption 2.2** (weak monotonicity of $r$). For all $\mu, \widetilde{\mu} \in \Delta(\mathcal{S})^H$, $\pi, \widetilde{\pi} \in (\Delta(\mathcal{A})^{\mathcal{S}})^H$, it holds that

$$\sum_{h=1}^H \sum_{(s,a) \in \mathcal{S} \times \mathcal{A}} (r_h(s, a, \mu_h) - r_h(s, a, \widetilde{\mu}_h))(\pi_h(a \mid s) \mu_h(s) - \widetilde{\pi}_h(a \mid s) \widetilde{\mu}_h(s)) \leq 0.$$

For example, a reward $r$ that satisfies these assumptions includes a model of a crowd that avoids overcrowding.

The second is the Lipschitz continuity of the reward $r$ with respect to $\mu \in (\Delta(\mathcal{S}))^H$, which is a standard assumption in the field of MFGs (Cui & Koeppl, 2021; Fabian et al., 2023; Zhang et al., 2023).

**Assumption 2.3** (Lipschitz continuity of $r$). There exists a constant $L$ such that for every $h \in [H]$, $s \in \mathcal{S}$, $a \in \mathcal{A}$, and $\mu, \mu' \in \Delta(\mathcal{S})$:

$$|r_h(s, a, \mu) - r_h(s, a, \mu')| \leq L\|\mu - \mu'\|.$$

Given a policy $\pi$, the probabilities $m[\pi] = (m[\pi]_h)_{h=1}^H \in \Delta(\mathcal{S})^H$ of the state is recursively defined as follows: $m[\pi]_1 = \mu_1$ and

$$m[\pi]_h(s_h) = \sum_{(s_{h-1}, a_{h-1}) \in \mathcal{S} \times \mathcal{A}} \pi_{h-1}(a_{h-1} \mid s_{h-1}) P_{h-1}(s_h \mid s_{h-1}, a_{h-1}) m[\pi]_{h-1}(s_{h-1}), \tag{2.1}$$

if $h = 2, \ldots, H$. We plan to maximize the following cumulative reward

$$J(\mu, \pi) := \sum_{h=1}^H \sum_{(s_h, a_h) \in \mathcal{S} \times \mathcal{A}} \pi_h(a_h \mid s_h) m[\pi]_h(s_h) r_h(s_h, a_h, \mu_h), \tag{2.2}$$

under a probability $\mu \in \Delta(\mathcal{S})^H$ of states. The *mean-field equilibrium* defined below means the pair of probabilities $\mu$ and policies $\pi$ that achieves the maximum under the constraints (2.1).

**Definition 2.4.** A pair $(\mu^\star, \pi^\star) \in \Delta(\mathcal{S})^H \times (\Delta(\mathcal{A})^{\mathcal{S}})^H$ is a *mean-field equilibrium* if it satisfies (i) $J(\mu^\star, \pi^\star) = \max_{\pi \in \Delta(\mathcal{S})^H} J(\mu^\star, \pi)$, and (ii) $\mu^\star = m[\pi^\star]$. In addition, set $\Pi^\star \subset (\Delta(\mathcal{A})^{\mathcal{S}})^H$ as the set of all policies that are in equilibrium.

Under Assumptions 2.2 and 2.3, there exists a mean-field equilibrium, see the proof of (Saldi et al., 2018, Theorem 3.3.) and (Pérolat et al., 2022, Proposition 1.). Note that the equilibrium may not be unique if the inequality in Assumption 2.2 is non-strict. In other words, the set $\Pi^\star \subset (\Delta(\mathcal{A})^{\mathcal{S}})^H$ is not singleton in general. As an illustrative example, one might consider the trivial case where $r \equiv 0$. Our goal is to construct an algorithm that generates policies that converge to $\Pi^\star$.

## 3 PROXIMAL POINT-TYPE METHOD FOR MFG

### 3.1 ALGORITHM

This section presents an algorithm motivated by the proximal point (PP) method. Let $\lambda > 0$ be a sufficiently small positive number, roughly "the inverse of learning rate." In the algorithm proposed in this paper, we generate a sequence $((\sigma^k, \mu^k))_{k=0}^\infty \subset (\Delta(\mathcal{A})^{\mathcal{S}})^H \times \Delta(\mathcal{S})^H$ as

$$\sigma^{k+1} = \underset{\pi \in (\Delta(\mathcal{A})^{\mathcal{S}})^H}{\arg\max} \left\{ J(\mu^{k+1}, \pi) - \lambda D_{m[\pi]}(\pi, \sigma^k) \right\}, \quad \mu^{k+1} = m[\sigma^{k+1}], \tag{3.1}$$

where $m$ is defined in (2.1) and $D_\mu(\pi, \sigma^k) := \sum_h \mathbb{E}_{s \sim \mu_h} [D_{\mathrm{KL}}(\pi_h(s), \sigma_h^k(s))]$ with a probability $\mu \in \Delta(\mathcal{S})^H$. If the initial policy $\pi^0$ has full support, i.e., $\min_{(h,s,a) \in H \times \mathcal{S} \times \mathcal{A}} \pi_h^0(a \mid s) > 0$, the rule (3.1) is well-defined, see Proposition C.1.

---

**Algorithm 1:** Proximal point (PP) method with KL divergence for MFG

---

**Input:** MFG $(\mathcal{S}, \mathcal{A}, H, P, r, \mu_1)$, initial policy $\pi^0$, number of iterations $N$, parameter $\lambda > 0$

**1 Initialization:** Set $k \leftarrow 0$ and $\sigma^k \leftarrow \pi^0$;

**2 while** $k < N$ **do**

**3**    Compute $(\mu^{k+1}, \sigma^{k+1})$ by solving the regularized MFG;

**4**

$$
\begin{cases}
\sigma^{k+1} = \underset{\pi \in (\Delta(\mathcal{A})^{\mathcal{S}})^H}{\arg\max} \left\{ J(\mu^{k+1}, \pi) - \lambda D_{m[\pi]}(\pi, \sigma^k) \right\}, \\
\mu^{k+1} = m[\sigma^{k+1}]
\end{cases}
$$

   Update $k \leftarrow k + 1$;

**Output:** $\sigma^k (\approx \pi^\star)$

---

Interestingly, the rule (3.1) is similar to the traditional proximal point (PP) method with KL divergence in mathematical optimization and Optimal Transport, see (Censor & Zenios, 1992; Xie et al., 2019) and the pseudocode in Algorithm 1. Therefore, we also refer to this update rule as the PP method. On the other hand, unlike the traditional PP method, our method changes the objective function $J(\mu^k, \bullet) \colon (\Delta(\mathcal{A})^{\mathcal{S}})^H \to \mathbb{R}$ with each iteration $k \in \mathbb{N}$. Therefore, the convergence of our traditional method is not directly derived from traditional theory.

## 3.2 LAST-ITERATE CONVERGENCE RESULT

The following theorem implies the last-iterate convergence of the policies generated by (3.1). Specifically, it shows that under the assumptions above, the sequence of policies converges to the equilibrium set. This result is crucial for the effectiveness of the algorithm in reaching an optimal policy.

**Theorem 3.1.** *Let $(\sigma^k)_{k=0}^\infty$ be the sequence defined by Algorithm 1. In addition to Assumptions 2.1 to 2.3, assume that the initial policy $\pi^0$ has full support, i.e., $\min_{(h,s,a) \in H \times \mathcal{S} \times \mathcal{A}} \pi_h^0(a \mid s) > 0$. Then, the sequence $(\sigma^k)_{k=0}^\infty$ converges to the set $\Pi^\star$ of equilibrium, i.e.,*

$$
\lim_{k \to \infty} \operatorname{dist}(\sigma^k, \Pi^\star) = 0,
$$

*where $\operatorname{dist}(\pi, \Pi^\star) \coloneqq \inf_{\pi^\star \in \Pi^\star} \sum_{(h,s) \in [H] \times \mathcal{S}} \|\pi_h(s) - \pi_h^\star(s)\|$.*

*Proof sketch of Theorem 3.1.* If we accept the next two lemmas, we can easily prove Theorem 3.1: The first implies that the KL divergence from an equilibrium to the generated policy becomes smaller as the cumulative reward $J$ increases.

**Lemma 3.2.** *Suppose Assumption 2.2. Then, for any equilibrium $(\mu^\star, \pi^\star)$ it holds that*

$$
D_{\mu^\star}(\pi^\star, \sigma^{k+1}) - D_{\mu^\star}(\pi^\star, \sigma^k) \leq J(\mu^\star, \sigma^{k+1}) - J(\mu^\star, \pi^\star).
$$

Furthermore, we can control the right-hand side of the inequality in Lemma 3.2 by the distance:

**Lemma 3.3.** *There exist positive constants $\alpha$ and $C$ such that, for any $\pi \in (\Delta(\mathcal{A})^{\mathcal{S}})^H$,*

$$
J(\mu^\star, \pi) - J(\mu^\star, \pi^\star) \leq -C(\operatorname{dist}(\pi, \Pi^\star))^\alpha.
$$

Combining these lemmas yields that $D_{\mu^\star}(\pi^\star, \sigma^{k+1}) - D_{\mu^\star}(\pi^\star, \sigma^k) \leq -C\big(\text{dist}(\sigma^{k+1}, \Pi^\star)\big)^\alpha$. Thus, the telescoping sum of this inequality yields

$$\sum_{k=1}^\infty \big(\text{dist}(\sigma^k, \Pi^\star)\big)^\alpha \leq \frac{1}{C} D_{\mu^\star}(\pi^\star, \pi^0) < +\infty.$$

Therefore, $\lim_{k\to\infty} \text{dist}(\sigma^k, \Pi^\star) = 0$. □

Thus, the non-trivial aspects of the last-iterate convergence lie in the proof of Lemmas 3.2 and 3.3; see Appendix B.

# 4 APPROXIMATING PROXIMAL POINT WITH MIRROR DESCENT IN REGULARIZED MFG

As in the previous section, in the PP method in Algorithm 1, it is necessary to solve the regularized MFG (3.1) at each iteration. Therefore, this section introduces Regularized Mirror Descent (RMD), which approximates the solution $(\mu^{k+1}, \sigma^{k+1})$ of (3.1) for each policy $\sigma^k$. The novel result in this section is that the divergence between the sequence of RMD and the equilibrium exponentially decays as shown in Figure 2.

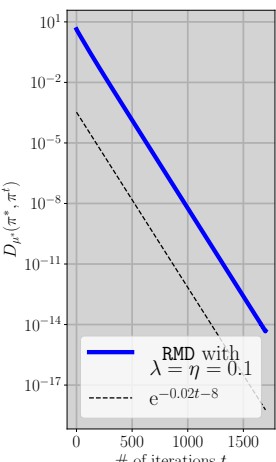

Figure 2: Behavior of RMD.

## 4.1 APPROXIMATION
OF THE UPDATE RULE OF PP WITH REGULARIZED MFG

Fortunately, solving (3.1) corresponds to finding an equilibrium for *KL-regularized MFG* introduced in Cui & Koeppl (2021); Zhang et al. (2023). Let us review the settings for the regularized MFG. For each parameter $\lambda > 0$ and policy $\sigma \in (\Delta(\mathcal{A})^{\mathcal{S}})^H$, which plays the role of $\sigma^k$ in Algorithm 1, we define the *regularized cumulative reward* $J^{\lambda,\sigma} \colon \Delta(\mathcal{S})^H \times (\Delta(\mathcal{A})^{\mathcal{S}})^H \ni (\mu, \pi) \mapsto J^{\lambda,\sigma}(\mu,\pi) \in \mathbb{R}$ to be

$$J^{\lambda,\sigma}(\mu,\pi) := J(\mu,\pi) - \lambda D_{m[\pi]}(\pi,\sigma). \tag{4.1}$$

Since $\sigma$ is a representative of $(\sigma^k)_k$, the assumption of full support is also imposed on $\sigma$:

**Assumption 4.1.** The base $\sigma$ has full support, i.e., $\sigma_{\min} := \min_{(s,a,h)\mathcal{S}\times\mathcal{A}\times[H]} \sigma_h(a \mid s) > 0$.

For the reward $J^{\lambda,\sigma}$, we introduce a *regularized equilibrium*:

**Definition 4.2.** A pair $(\mu^*, \varpi^*) \in \Delta(\mathcal{S})^H \times (\Delta(\mathcal{A})^{\mathcal{S}})^H$ is *regularized equilibrium* of $J^{\lambda,\sigma}$ if it satisfies (i) $J^{\lambda,\sigma}(\mu^*, \varpi^*) = \max_{\pi \in \Delta(\mathcal{S})^H} J^{\lambda,\sigma}(\mu^*, \pi)$, and (ii) $\mu^* = m[\varpi^*]$.

Specifically, $(\mu^{k+1}, \sigma^{k+1})$ can be characterized as the regularized equilibrium of $J^{\lambda,\sigma^k}$ for $k \in \mathbb{N}$. Note that the regularized equilibrium is unique under Assumption 4.1, see Appendix C.

In the next subsection, we will introduce RMD using *value functions*, which are defined as follows: for each $h \in [H]$, $s \in \mathcal{S}$, $a \in \mathcal{A}$, $\mu \in \Delta(\mathcal{S})^H$ and $\pi \in \Delta(\mathcal{A})^{\mathcal{S}}$, define the *state value function* $V_h^{\lambda,\sigma} \colon \mathcal{S} \times \Delta(\mathcal{S})^H \times (\Delta(\mathcal{A})^{\mathcal{S}})^H \to \mathbb{R}$ and the *state-action value function* $Q_h^{\lambda,\sigma} \colon \mathcal{S} \times \mathcal{A} \times \Delta(\mathcal{S})^H \times (\Delta(\mathcal{A})^{\mathcal{S}})^H \to \mathbb{R}$ as

$$V_h^{\lambda,\sigma}(s,\mu,\pi) := \mathbb{E}_{((s_l,a_l))_{l=h}^H} \left[ \sum_{l=h}^H \left( r_l(s_l,a_l,\mu_l) - \lambda D_{\mathrm{KL}}(\pi_l(s_l), \sigma_l(s_l)) \right) \,\middle|\, s_h = s \right], \tag{4.2}$$

$$V_{H+1}^{\lambda,\sigma}(s,\mu,\pi) := 0,$$

$$Q_h^{\lambda,\sigma}(s,a,\mu,\pi) = r_h(s,a,\mu_h) + \mathbb{E}_{s_{h+1}\sim P(s,a,\mu_h)} \left[ V_{h+1}^{\lambda,\sigma}(s_{h+1},\mu,\pi) \right]. \tag{4.3}$$

Here, the discrete time stochastic process $((s_l,a_l))_{l=h}^H$ is induced recursively as $s_{l+1} \sim P_l(s_l,a_l), a_l \sim \pi_l(s_l)$ for each $l \in \{h, \ldots, H-1\}$ and $a_H \sim \pi_H(s_H)$. Note that the the objective function $J^{\lambda,\sigma}$ in Definition 4.2 can be expressed as $J^{\lambda,\sigma}(\mu,\pi) = \mathbb{E}_{s\sim\mu_1}[V_1^{\lambda,\sigma}(s,\mu,\pi)]$.

---

**Algorithm 2:** Practical version of Algorithm 1 for MFG

---

**Input:** MFG$(\mathcal{S}, \mathcal{A}, H, P, r, \mu_1)$, initial policy $\pi^0$, number of iterations $N$, parameter $\lambda > 0$

**1 Initialization:** Set $k \leftarrow 0$ and $\sigma^k \leftarrow \pi^0$;

**2 while** $k < N$ **do**

**3**     Compute $(\mu^{k+1}, \sigma^{k+1})$ by solving the regularized MFG;

**4**

$$
\begin{cases}
\sigma^{k+1} = \texttt{RMD}(\text{MFG}, \sigma^k, \lambda, \eta, \sigma^k, \tau), \\
\mu^{k+1} = m[\sigma^{k+1}]
\end{cases}
$$

        Update $k \leftarrow k + 1$;

    **Output:** $\sigma^k (\approx \pi^\star)$

**5**

---

**6 Function** $\texttt{RMD}(\text{MFG}, \pi^0, \lambda, \eta, \sigma^0, \tau)$:

**7**     **Initialization:** Set $t \leftarrow 0$, $\pi^t \leftarrow \pi^0$ and $\sigma \leftarrow \sigma^0$;

**8**     **while** $t < \tau$ **do**

**9**         Compute $\mu^t = m[\pi^t]$;

**10**         Compute $Q_h^{\lambda,\sigma}(s, a, \pi^t, \mu^t)$ $((h, s, a) \in [H] \times \mathcal{S} \times \mathcal{A})$ by (4.3);

**11**         Compute $\pi^{t+1}$ as, for $(h, s, a) \in [H] \times \mathcal{S} \times \mathcal{A}$,

$$
\pi_h^{t+1}(a \mid s) = \frac{(\sigma_h(a \mid s))^{\lambda\eta}(\pi_h^t(a \mid s))^{1-\lambda\eta} \exp\left(\eta Q_h^{\lambda,\sigma}(s, a, \pi^t, \mu^t)\right)}{\sum\limits_{a' \in \mathcal{A}} (\sigma_h(a' \mid s))^{\lambda\eta}(\pi_h^t(a' \mid s))^{1-\lambda\eta} \exp\left(\eta Q_h^{\lambda,\sigma}(s, a', \pi^t, \mu^t)\right)}
$$

            Update $t \leftarrow t + 1$;

**12**     **return** $\pi^t$;

---

### 4.2 An exponential convergence result of Regularized Mirror Descent

In this subsection, we introduce the iterative method for finding the regularized equilibrium proposed by Zhang et al. (2023) as RMD. The method constructs a sequence $\left((\pi^t, \mu^t)\right)_{t=0}^{\infty} \subset (\Delta(\mathcal{A})^{\mathcal{S}})^H \times \Delta(\mathcal{S})^H$ approximating the regularized equilibrium of $J^{\lambda,\sigma}$ using the following rule:

$$
\begin{cases}
\pi_h^{t+1}(s) = \underset{p \in \Delta(\mathcal{A})}{\arg\max} \left\{ \frac{\eta}{1-\lambda\eta} \left( \left\langle Q_h^{\lambda,\sigma}(s, \bullet, \pi^t, \mu^t), p \right\rangle - \lambda D_{\mathrm{KL}}(p, \sigma_h(s)) \right) - D_{\mathrm{KL}}(p, \pi_h^t(s)) \right\}, \\
\mu^{t+1} = m[\pi^{t+1}],
\end{cases}
$$

where $\eta > 0$ is another learning rate, and $Q_h^{\lambda,\sigma}$ is the state-action value function defined in (4.3). We give the pseudo-code of RMD in Algorithm 2. For the sequence of policies in RMD, we can establish the convergence result as follows:

**Theorem 4.3.** *Let $((\mu^t, \pi^t))_{t=0}^{\infty} \subset \Delta(\mathcal{S})^H \times (\Delta(\mathcal{A})^{\mathcal{S}})^H$ be the sequence generated by (4.4), and $(\mu^*, \varpi^*) \in \Delta(\mathcal{S})^H \times (\Delta(\mathcal{A})^{\mathcal{S}})^H$ be the regularized equilibrium given in Definition 4.2. In addition to Assumptions 2.2, 2.3, and 4.1, suppose that $\eta \leq \eta^*$, where $\eta^* > 0$ is the upper bound of the learning rate defined in (D.5), which only depends on $\lambda$, $\sigma$, $H$ and $|\mathcal{A}|$. Then, the sequence $(\pi^t)_{t=0}^{\infty}$ satisfies*

$$
D_{\mu^*}(\varpi^*, \pi^{t+1}) \leq \left(1 - \frac{\lambda\eta}{2}\right) D_{\mu^*}(\varpi^*, \pi^t) \quad (t = 0, 1, \dots).
$$

*Accordingly, $D_{\mu^*}(\varpi^*, \pi^t) \leq D_{\mu^*}(\varpi^*, \pi^0) \exp\left(-\lambda\eta t/2\right)$. Clearly, the inequality states that an approximate policy $\pi^t$ satisfying $D_{\mu^*}(\varpi^*, \pi^t) < \varepsilon$ can be obtained in $\mathcal{O}(\log(1/\varepsilon))$ iterations.*

### 4.3 INTUITION FOR EXPONENTIAL CONVERGENCE: CONTINUOUS-TIME VERSION OF REGULARIZED MIRROR DESCENT

The convergence of $(\pi^t)_{t=0}^{\infty}$ can be intuitively explained by considering a continuous limit $(\pi^t)_{t\geq 0}$ with respect to the time $t$ of RMD. In this paragraph, we will use the idea of mirror flow (Krichene et al., 2015; Tzen et al., 2023; Deb et al., 2023) and continuous dynamics in games (Taylor & Jonker, 1978; Mertikopoulos et al., 2018; Pérolat et al., 2021; 2022) to observe the exponential convergence of the flow to equilibrium. According to Deb et al. (2023, (2.1)), the continuous curve of $\pi$ should satisfy that

$$\frac{\mathrm{d}}{\mathrm{d}t}\pi_h^t\left(a \mid s\right) = \pi_h^t\left(a \mid s\right)\left(Q_h^{\lambda,\sigma}(s,a,\pi^t,\mu^t) - \lambda\log\frac{\pi_h^t\left(a \mid s\right)}{\sigma_h\left(a \mid s\right)}\right). \tag{4.4}$$

The flow induced by the dynamical system (4.4) converges to equilibrium *exponentially* as time $t$ goes to infinity.

> **Theorem 4.4.** *Let $\pi^t$ be a solution of (4.4) and $\varpi^*$ be a regularized equilibrium defined in Definition 4.2. Suppose that Assumption 2.2. Then, it holds that*
>
> $$\frac{\mathrm{d}}{\mathrm{d}t}D_{\mu^*}(\varpi^*,\pi^t) \leq -\lambda D_{\mu^*}(\varpi^*,\pi^t),$$
>
> *for all $t \geq 0$. Moreover, the inequality implies $D_{\mu^*}(\varpi^*,\pi^t) \leq D_{\mu^*}(\varpi^*,\pi^0)\exp\left(-\lambda t\right)$.*

Technically, the non-Lipschitz continuity of the value function $Q_h^{\lambda,\sigma}(s,a,\bullet,\mu^t)$ in the right-hand side of (4.4) is non-trivial for the existence of the solution $\pi\colon[0,+\infty) \to (\Delta(\mathcal{A})^{\mathcal{S}})^H$ of the differential equation (4.4), see, e.g., (Coddington & Levinson, 1984). The proof of this existence and Theorem 4.4 are given in Appendix C.

### 4.4 PROOF SKETCH OF THE CONVERGENCE RESULT FOR REGULARIZED MIRROR DESCENT

Let us return from continuous-time dynamics (4.4) to the discrete-time algorithm (4.4). The technical difficulty in the proof of Theorem 4.3 is the non-Lipschitz continuity of the value function $Q_h^{\lambda,\sigma}$ in (4.4), that is, the derivative of $Q_h^{\lambda,\sigma}(s,a,\pi,\mu)$ with respect to the policy $\pi$ can blow up as $\pi$ approaches the boundary of the space $(\Delta(\mathcal{A})^{\mathcal{S}})^H$ of probability simplices.

We can overcome this difficulty as shown in the following sketch of proof:

> ***Proof sketch of Theorem 4.3.*** In a similar way to Theorem 4.4, we can obtain the following inequality with a discretization error:
>
> $$D_{\mu^*}(\varpi^*,\pi^{t+1}) - D_{\mu^*}(\varpi^*,\pi^t) \leq -\lambda\eta D_{\mu^*}(\varpi^*,\pi^t) + \boxed{D_{\mu^*}(\pi^t,\pi^{t+1}).} \tag{4.5}$$
>
> discretization error
>
> The remainder of the proof is almost entirely dedicated to showing that the above error term is sufficiently small and bounded compared to the other terms in the inequality. As a result, we obtain the following claim:

> **Claim 4.5.** *Suppose that the learning rate $\eta$ is less than the upper bound $\eta^*$ in (D.5). Then, we have*
>
> $$\boxed{D_{\mu^*}(\pi^t,\pi^{t+1})} \leq C\eta^2 D_{\mu^*}(\varpi^*,\pi^t),$$
>
> *where $C > 0$ is the constant defined in (D.4), which satisfies $C\eta^* \leq \lambda/2$.*

> The key to proving Claim 4.5 is leveraging another claim that, over the sequence $(\pi^t)_t$, the value function $Q_h^{\lambda,\sigma}$ behaves well, almost as if it were a Lipschitz continuous function, see Lemma D.3 for details. Therefore, applying Claim 4.5 to (4.5) completes the proof. $\qquad\square$

The complete proof of Theorem 4.3 is given in Appendix D.

### 4.5 APPROXIMATED PROXIMAL POINT METHOD

Let us consider an approximation of Algorithm 1 using RMD of (4.4). We can simply replace the intractable computation in line 4 of Algorithm 1 with RMD. In the end, this means that after repeating (4.4) a sufficient number of times, we also update $\sigma$ to the most recently obtained policy $\sigma^{k+1}$ using RMD. The pseudo-code that summarizes this idea is presented in Algorithm 2.

## 5 NUMERICAL EXPERIMENT

We numerically demonstrate that the proposed algorithm (Algorithm 2), which is the approximated version of Algorithm 1, can achieve convergence to the mean-field equilibrium.

**Algorithms.** In this experiment, we implement Algorithm 2. For comparison, we also implement RMD (i.e., Algorithm 2 without the update of $\sigma_k$) in (4.4). For both algorithms, the learning rate is fixed at $\eta = 0.1$, and we vary the regularization parameter $\lambda$ and update time $T$ to run the experiments.

**Evaluations.** We evaluate the convergence of our proposed method using the Beach Bar Process introduced by Perrin et al. (2020), a standard benchmark for MFGs. In particular, the transition kernel $P$ in this benchmark gives a random walk on a one-dimensional discretized torus $\mathcal{S} = \{0, \ldots, |\mathcal{S}| - 1\}$, and the reward is set to be $r_h(s, a, \mu) = -|a|/|\mathcal{S}| - |s - |\mathcal{S}|/2|/|\mathcal{S}| - \log \mu_h(s)$ with $a \in \mathcal{A} := \{-1, \pm 0, +1\}$. See Appendix F for further details. Since the mean-field equilibrium in this benchmark cannot be computed exactly, we follow Pérolat et al. (2022); Zhang et al. (2023) and employ the exploitability of a policy $\pi \in (\Delta(\mathcal{A})^{\mathcal{S}})^H$ defined by

$$\text{Exploit}(\pi) := \max_{\pi' \in (\Delta(\mathcal{A})^{\mathcal{S}})^H} \{J(m[\pi], \pi')\} - J(m[\pi], \pi) \geq 0,$$

as our convergence criterion. Note that from Definition 2.4, $\text{Exploit}(\pi) = 0$ if and only if $(m[\pi], \pi)$ is mean-field equilibrium.

**Discussion.** Figure 3 is a summary of the results of the experiment. The most noteworthy aspect is the convergence of the exploitability, as shown in Figure 3b. Our proposed method decreases the exploitability with each iteration when we update $\sigma$.

Figures 3a and 3c illustrate the qualitative validity of the approximation achieved by our proposed method. In this benchmark, the equilibrium is expected to lie at the vertices of the probability simplex. Therefore, RMD, which can shift the equilibrium to the interior of the probability simplex, seems unable to find the mean-field equilibrium accurately. On the other hand, the sequence $(\pi^t)_t$ of policies generated by our proposed method shows a behavior that converges to the vertices.

In summary, Algorithm 2 experimentally shows the last-iterate convergence to the mean-field equilibrium. This is evidenced by the decreasing exploitability and the qualitative behavior in our proposed method, which align with the theoretical guarantees.

## 6 COMPARISON OF THE RESULTS

**Last-iterate convergence (LIC) results for MFG.** Pérolat et al. (2022) showed that Mirror Descent achieves LIC only under *strictly* monotone conditions, i.e., if the equality in the Lemma E.2 is satisfied only if $\pi = \widetilde{\pi}$. In contrast, our work establishes LIC even in *non-strictly* monotone scenarios. While the distinction regarding strictness might seem subtle, it is profoundly significant. Indeed, non-strictly monotone MFGs encompass the fundamental examples of finite-horizon Markov Decision Processes. Moreover, in strictly monotone cases, mean-field equilibria become unique. Consequently, as Zeng et al. (2024) also noted, strictly monotone rewards fail to represent MFGs with diverse equilibria.

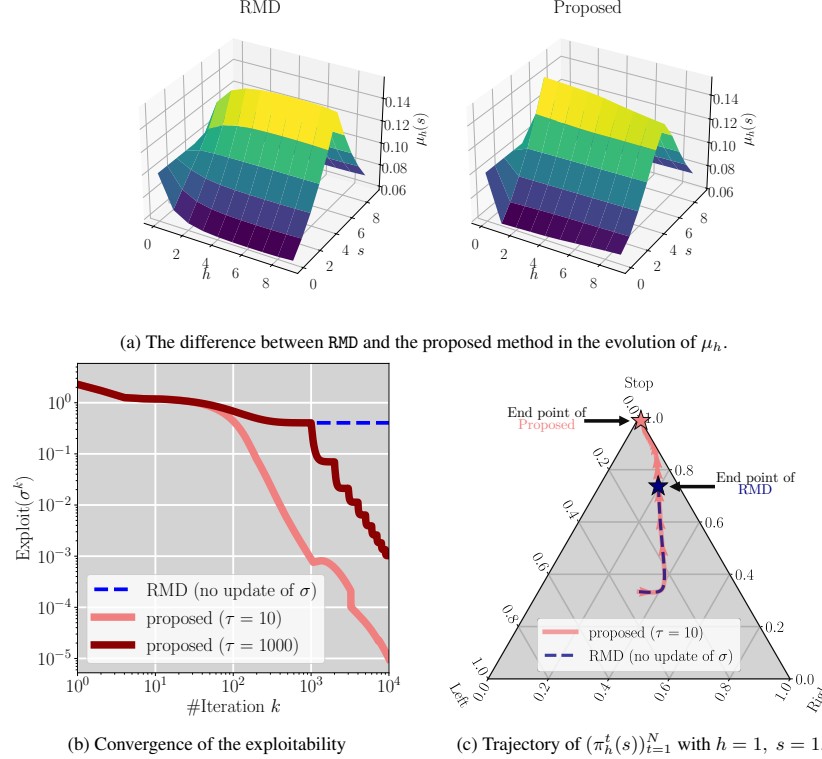

(a) The difference between RMD and the proposed method in the evolution of $\mu_h$.

(b) Convergence of the exploitability

(c) Trajectory of $(\pi_h^t(s))_{t=1}^N$ with $h = 1$, $s = 1$.

Figure 3: Experimental results for Algorithm 2 for Beach Bar Process

**Regularized MFG.** Theorem 4.3, which supports the efficient execution of RMD, is novel in two respects: RMD achieves LIC, and the divergence to the equilibrium decays exponentially. Indeed, one of the few works that analyze the convergence rate of RMD states that the time-averaged policy $\frac{1}{T} \sum_{t=0}^{T} \pi^t$ up to time $T$ converges to the equilibrium in $\mathcal{O}(1/\varepsilon^2)$ iterations (Zhang et al., 2023). Additionally, although it is a different approach from MD, it is known that applying fixed-point iteration to regularized MFG achieves an exponential convergence rate under the assumption that the regularization parameter $\lambda$ is sufficiently large (Cui & Koeppl, 2021). In contrast, our work derives the convergence rate for cases where $\lambda$ is sufficiently small.

**Other type of learning method of MFG.** Recently, in addition to Mirror Descent and Fictitious Play, a new type of learning method using the characterization of MFGs as optimization problems has been proposed (Guo et al., 2024; Hu & Zhang, 2024). In this work, the authors establish local convergence of the algorithms without the assumption of monotonicity. Specifically, it is proved that an optimization method can achieve LIC if the initial guess of the algorithm is sufficiently close to the Nash equilibrium. In contrast, our convergence results state "global" convergence under the assumption of monotonicity, complementing their results. See Table 1 in the Appendix for a comparison of our results with the more comprehensive previous studies.

## 7 CONCLUSION

This paper proposes noble algorithms that can achieve last-iterate convergence under the monotonicity condition. The main idea behind the derivation of the main algorithm (Algorithm 2) is to approximate the proximal-point type algorithm (Algorithm 1) using RMD. Theorem 3.1 guarantees that the proximal-point-type algorithm achieves LIC, and Theorem 4.3 guarantees the exponential convergence of RMD. An important future task of this study is to prove the convergence rates of Algorithm 2. Specifically, we aim to make the convergence result of Theorem 3.1 quantitative. As the experimental results suggest in Figure 3b, we conjecture that the algorithm converges with a rate of $\mathcal{O}(1/t^\alpha)$ for some $\alpha > 0$.

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

# A RELATED WORKS

Several previous studies have derived convergence results for MFG for various algorithms. Note that the meaning of convergence is different in the previous studies. Table 1 shows which type of convergence is obtained for which type of algorithm in the previous studies and our results. Based on Table 1, we discuss the technical contributions of this paper below:

**Significance of Our Convergence Results:** Unlike many of the referenced works that require strong assumptions, such as contraction, to achieve convergence, our results demonstrate last-iterate convergence (LIC) without such stringent conditions. This highlights that our contributions fill a significant gap in the literature.

**LIC of RMD:** Achieving exponential convergence rates to the regularized equilibrium is challenging with existing techniques. Our technical contributions include deriving discretization errors in Equation (4.5) that are distinct from those in policy optimization by Zhan et al. (2021) and regularized MFG (Zhang et al., 2023).

**The Difficulty of Applying the Three-Point Lemma to MFGs:** The three-point lemma in (Zhan et al., 2021, Lemma 6) cannot be directly applied to MFGs. The main reason is that the inner product $\langle Q^k(s), \pi^{k+1}(s) - p \rangle$ in the right-hand side of the three-point lemma concerns the policy at iteration index $k+1$, not $k$. In our analysis (as shown on page 18), this term is transformed into $\langle Q^k(s), \pi^k(s) - p \rangle$, which allows us to apply a crucial lemma (Lemma E.4) that holds for MFGs. This transformation is non-trivial and essential for our analysis. In the three-point lemma, the term $D_{h_s}(\pi^{(k+1)}, \pi^{(k)})$ appears as a discretization error. In contrast, our analysis derives a reverse version $D_{\mu^*}(\pi^k, \pi^{k+1})$. This distinction is significant, especially for non-symmetric divergences such as the KL divergence. The reverse order in our analysis is crucial for the theoretical guarantees we provide.

# B PROOF OF THEOREM 3.1

**Proof of Lemma 3.2.** Let $(\mu^\star, \pi^\star)$ be a mean-field equilibrium defined in Definition 2.4. By the update rule (3.1) and Lemma E.1, we have

$$\left\langle Q_h^{\lambda, \sigma^k}(s, \bullet, \sigma^{k+1}, \mu^{k+1}) - \lambda \log \frac{\sigma_h^{k+1}(s)}{\sigma_h^k(s)}, (\pi_h^\star - \sigma_h^{k+1})(s) \right\rangle \leq 0,$$

Table 1: Related work of convergence in MFGs

| | **Learning algorithm** | **Summary of convergence results** |
|---|---|---|
| Xie et al. (2021) | Fictious play | time-averaging convergence |
| Zhang et al. (2023) | RMD | time-averaging convergence (to regularized equilibrium) under monotonicity |
| Mao et al. (2022) | Actor-critic | time-averaging convergence (to regularized equilibrium) |
| Zeman et al. (2023) | Q-learning | time-averaging convergence |
| Yardim et al. (2023) | Mirror Descent | LIC under contraction |
| Zeng et al. (2024) | Actor-critic | best-iterate convergence under Herding |
| Huang et al. (2024) | Maximum Likelihood Estimation | N/A |
| Angiuli et al. (2022) | (two-time scale) Q-Learning | N/A |
| Angiuli et al. (2024) | (three-time scale) Q-learning | LIC under contraction |
| Pérolat et al. (2021) | RMD | LIC (to regularized equilibrium) under strict monotonicity in continuous-time |
| **Our work (Theorem 3.1)** | Proximal Point | LIC under monotonicity |
| **Our work (Theorem 4.4)** | RMD | LIC (to regularized equilibrium) under monotonicity |

for each $h \in [H]$, $s \in \mathcal{S}$ and $k \in \mathbb{N}$, i.e.,

$$D_{\mathrm{KL}}(\pi_h^\star(s), \sigma_h^{k+1}(s)) - D_{\mathrm{KL}}(\pi_h^\star(s), \sigma_h^k(s)) - D_{\mathrm{KL}}(\sigma_h^{k+1}(s), \sigma_h^k(s))$$
$$\leq \frac{1}{\lambda} \left\langle Q_h^{\lambda, \sigma^k}(s, \bullet, \sigma^{k+1}, \mu^{k+1}), (\sigma_h^{k+1}) - \pi_h^\star)(s) \right\rangle . \tag{B.1}$$

Taking the expectation with respect to $s \sim \mu_h^\star$ and summing (B.1) over $h \in [H]$ yields

$$D_{\mu^\star}(\pi^\star, \sigma^{k+1}) - D_{\mu^\star}(\pi^\star, \sigma^k) + D_{\mu^\star}(\sigma^{k+1}, \sigma^k)$$
$$\leq \frac{1}{\lambda} \sum_{h=1}^{H} \mathbb{E}_{s \sim \mu_h^\star} \left[ \left\langle Q_h^{\lambda, \sigma^k}(s, \bullet, \sigma^{k+1}, \mu^{k+1}), (\sigma_h^{k+1}) - \pi_h^\star)(s) \right\rangle \right].$$

By virtue of Lemmas E.2 and E.4, we further have

$$\sum_{h=1}^{H} \mathbb{E}_{s \sim \mu_h^\star} \left[ \left\langle Q_h^{\lambda, \sigma^k}(s, \bullet, \sigma^{k+1}, \mu^{k+1}), (\sigma_h^{k+1}) - \pi_h^\star)(s) \right\rangle \right]$$
$$\leq J^{\lambda, \sigma^k}(\mu^{k+1}, \sigma^{k+1}) - J^{\lambda, \sigma^k}(\mu^{k+1}, \pi^\star) - \lambda D_{\mu^\star}(\pi^\star, \sigma^k) + \lambda D_{\mu^\star}(\sigma^{k+1}, \sigma^k)$$
$$\leq J^{\lambda, \sigma^k}(\mu^\star, \sigma^{k+1}) - J^{\lambda, \sigma^k}(\mu^\star, \pi^\star) - \lambda D_{\mu^\star}(\pi^\star, \sigma^k) + \lambda D_{\mu^\star}(\sigma^{k+1}, \sigma^k)$$
$$\leq J(\mu^\star, \sigma^{k+1}) - J(\mu^\star, \pi^\star) - \lambda D_{\mu^{k+1}}(\sigma^{k+1}, \sigma^k) + \lambda D_{\mu^\star}(\sigma^{k+1}, \sigma^k),$$

where we use the identity $J^{\lambda, \sigma^k}(\mu^\star, \pi) = J(\mu^\star, \pi) - \lambda D_{m[\pi]}(\pi, \sigma^k)$ for $\pi \in (\Delta(\mathcal{A})^\mathcal{S})^H$, and Definition 2.4. ∎

**Proof of Lemma 3.3.** Note that the function $J(\mu^\star, \bullet)\colon (\Delta(\mathcal{A})^{\mathcal{S}})^H \ni \pi \mapsto J(\mu^\star, \pi) \in \mathbb{R}$ is real-analytic. Therefore, we can apply (Łojasiewicz, 1971, §18, Théorème 2.). ∎

## C   PROOF OF THEOREM 4.4

**Proof of Theorem 4.4.** Let $h^\star\colon \mathbb{R}^{|\mathcal{A}|} \to \mathbb{R}$ be the convex conjugate of $h$, i.e., $h^\star(y) = \sum_{a \in \mathcal{A}} \exp(y(a))$ for $y \in \mathbb{R}^{|\mathcal{A}|}$. From direct computations, we have

$$
\frac{\mathrm{d}}{\mathrm{d}t} D_{\mu^*}(\varpi^*, \pi^t)
$$

$$
= \sum_{h=1}^{H} \mathbb{E}_{s \sim \mu_h^*} \left[ \frac{\mathrm{d}}{\mathrm{d}t} D_{\mathrm{KL}}(\varpi_h^*(s), \pi^t(s)) \right]
$$

$$
= \sum_{h=1}^{H} \mathbb{E}_{s \sim \mu_h^*} \left[ \left\langle 1 - \frac{\varpi_h^*(s)}{\pi_h^t(s)}, \frac{\mathrm{d}}{\mathrm{d}t} \pi_h^t(s) \right\rangle \right]
$$

$$
= \sum_{h=1}^{H} \mathbb{E}_{s \sim \mu_h^*} \left[ \left\langle 1 - \frac{\varpi_h^*(s)}{\pi_h^t(s)}, \pi_h^t(a \mid s) \left( Q_h^{\lambda,\sigma}(s, a, \pi^t, \mu^t) - \lambda \log \frac{\pi_h^t(a \mid s)}{\sigma_h(a \mid s)} \right) \right\rangle \right]
$$

$$
= \sum_{h=1}^{H} \mathbb{E}_{s \sim \mu_h^*} \left[ \left\langle (\pi_h^t - \varpi_h^*)(s), Q_h^{\lambda,\sigma}(s, \bullet, \pi^t, \mu^t) - \lambda \log \frac{\pi_h^t(a \mid s)}{\sigma_h(a \mid s)} \right\rangle \right]
$$

$$
= \sum_{h=1}^{H} \mathbb{E}_{s \sim \mu_h^*} \left[ \left\langle (\pi_h^t - \varpi_h^*)(s), Q_h^{\lambda,\sigma}(s, \bullet, \pi^t, \mu^t) \right\rangle \right] - \lambda \sum_{h=1}^{H} \mathbb{E}_{s \sim \mu_h^*} \left[ \left\langle (\pi_h^t - \varpi_h^*)(s), \log \frac{\pi_h^t(s)}{\sigma_h(s)} \right\rangle \right].
$$

We apply Lemma E.4 for the first term and get

$$
\sum_{h=1}^{H} \mathbb{E}_{s \sim \mu_h^*} \left[ \left\langle (\pi_h^t - \varpi_h^*)(s), Q_h^{\lambda,\sigma}(s, \bullet, \pi^t, \mu^t) \right\rangle \right]
$$
$$
= J^{\lambda,\sigma}(\mu^t, \pi^t) - J^{\lambda,\sigma}(\mu^t, \varpi^*) - \lambda D_{\mu^*}(\varpi^*, \sigma) + \lambda D_{\mu^*}(\pi^t, \sigma). \tag{C.1}
$$

Similarly, we apply Lemma E.5 for the second term and get

$$
\sum_{h=1}^{H} \mathbb{E}_{s \sim \mu_h^*} \left[ \left\langle (\pi_h^t - \varpi_h^*)(s), \log \frac{\pi_h^t(s)}{\sigma_h(s)} \right\rangle \right] = D_{\mu^*}(\pi^t, \sigma) - D_{\mu^*}(\varpi^*, \sigma) + D_{\mu^*}(\varpi^*, \pi^t). \tag{C.2}
$$

Combining (C.1) and (C.2) yields

$$
\frac{\mathrm{d}}{\mathrm{d}t} D_{\mu^*}(\varpi^*, \pi^t) = J^{\lambda,\sigma}(\mu^t, \pi^t) - J^{\lambda,\sigma}(\mu^t, \varpi^*) - \lambda D_{\mu^*}(\varpi^*, \pi^t).
$$

By virtue of the definition of mean-field equilibrium and Lemma E.2, we find

$$
J^{\lambda,\sigma}(\mu^t, \pi^t) - J^{\lambda,\sigma}(\mu^t, \varpi^*) \leq J^{\lambda,\sigma}(\mu^*, \pi^t) - J^{\lambda,\sigma}(\mu^*, \varpi^*) \leq 0.
$$

Therefore, we obtain

$$
\frac{\mathrm{d}}{\mathrm{d}t} D_{\mu^*}(\varpi^*, \pi^t) \leq -\lambda D_{\mu^*}(\varpi^*, \pi^t).
$$

∎

**Proposition C.1.** *Assume the same assumption as in Theorem 3.1. Then, there exists a unique maximizer of $J^{\lambda,\sigma^k}(\mu^k, \bullet)\colon (\Delta(\mathcal{A})^{\mathcal{S}})^H \to \mathbb{R}$ for each $k \in \mathbb{N}$.*

The uniqueness of Proposition C.1 is a new result. The proof uses a continuous-time dynamics shown in Theorem 4.4, see Appendix C. In the following proof, we employ the same proof strategy as in (Chill et al., 2010, Theorem 2.10). Before the proof, set $v_{s,h}^{\lambda,\sigma}(\pi) :=$ $\pi_h(a \mid s)\left(Q_h^{\lambda,\sigma}(s, a, \pi, m[\pi]) - \lambda \log \frac{\pi_h(a \mid s)}{\sigma_h(a \mid s)}\right)$ for $\pi \in (\Delta(\mathcal{A})^{\mathcal{S}})^H$.

***Proof of Proposition C.1.*** The existence is shown by a slightly modified version of (Zhang et al., 2023, Theorem 2). It remains to prove the uniqueness. Fix the regularized equilibrium $\varpi^* \in (\Delta(\mathcal{A})^{\mathcal{S}})^H$.

First of all, we prove the global existence of (4.4). By the local Lipschitz continuity of the right-hand side of the dynamics (4.4) and Picard–Lindelöf theorem, there exists a unique maximal solution $\pi$ of (4.4) with the initial condition $\pi|_{t=0} = \pi^0$. Namely, there exist $T \in (0, +\infty]$ and $\pi \colon [0, T) \to \mathbb{R}^{|\mathcal{A}|}$ such that $\pi$ is differentiable on $(0, T)$ and it holds that (4.4) for all $t \in (0, T)$. Thus, Theorem 4.4 ensures that

$$D_{\mu^*}(\varpi^*, \pi^t) + \lambda \int_0^t D_{\mu^*}(\varpi^*, \pi^\tau)\, \mathrm{d}\tau \le D_{\mu^*}(\varpi^*, \pi^0) =: c < +\infty,$$

for every $t \in [0, T)$. As a result, the trajectory $\left\{\pi^t \in (\Delta(\mathcal{A})^{\mathcal{S}})^H \mid t \in [0, T)\right\}$ is included in $K_c :=$ $\left\{\pi \in (\Delta(\mathcal{A})^{\mathcal{S}})^H \mid D_{\mu^*}(\varpi^*, \pi) \le c\right\}$. Note that $K_c$ is compact from Pinsker inequality.

Since the right-hand side of (4.4) is continuous on $K_c$, we obtain $\sup_{t \in [0, +\infty)} \left\|v_{s,h}^{\lambda,\sigma}(\pi^t)\right\| < +\infty$. Thus, the equation (4.4) implies $\left\|\frac{\mathrm{d}\pi^t}{\mathrm{d}t}\right\|$ is uniformly bounded on $[0, T)$. Hence, $\pi$ extends to a continuous function on $[0, T]$.

To obtain a contradiction, we assume $T < +\infty$. Then, there exists the solution $\pi'$ of (4.4) on a larger interval than $\pi$ with a new initial condition $\pi'|_{t'=T} = \pi^T$, which contradicts the maximality of the solution $\pi$.

Therefore, the limit $\lim_{t \to \infty} \pi^t$ exists and is equal to $\varpi^*$. Here, $\varpi^*$ is arbitrary, so the regularized equilibrium is unique. ∎

# D   PROOF OF THEOREM 4.3

**Lemma D.1.** *It holds that*

$$\left\langle \eta\left(Q_h^{\lambda,\sigma}(s, \bullet, \pi^t, \mu^t) - \lambda \log \frac{\pi_h^{t+1}(s)}{\sigma_h(s)}\right) - (1 - \lambda\eta) \log \frac{\pi_h^{t+1}(s)}{\pi_h^t(s)}, \delta \right\rangle = 0,$$

*for all $\delta \in \mathbb{R}^{|\mathcal{A}|}$ such that $\sum_a \delta(a) = 0$.*

We introduce the following lemma:

**Lemma D.2.** *Let $(\pi^t)_t$ be the sequence defined by (4.4) and $\varpi^*$ be the policy satisfies Definition 4.2. Assume that there exist vectors $w_h^\sigma$ and $w_h^0(s) \in \mathbb{R}^{|\mathcal{A}|}$ satisfying*

$$\lambda H \log \sigma_{\min} \le w_h^\sigma(a \mid s) \le -\lambda H \log \sigma_{\min}, \qquad \sigma_h(a \mid s) \propto \exp\left(\frac{w_h^\sigma(a \mid s)}{\lambda}\right),$$

$$2\lambda H \log \sigma_{\min} \le w_h^0(a \mid s) \le H, \qquad \pi_h^0(a \mid s) \propto \exp\left(\frac{w_h^0(a \mid s)}{\lambda}\right).$$

*for all $a \in \mathcal{A}.\pi^0 \in (\Delta(\mathcal{A})^{\mathcal{S}})^H$, $h \in [H]$ and $s \in \mathcal{S}$. Then, for any $h \in [H], s \in \mathcal{S}$, and $t \ge 0$, it holds that*

$$\max\left\{\left\|\log \pi_h^t(s)\right\|_\infty, \left\|\log \pi_h^*(s)\right\|_\infty\right\} \le \frac{H(1 - \lambda \log \sigma_{\min})}{\lambda} + \log|\mathcal{A}|.$$

**Proof.** We first show that $\pi_h^t$ can be written as

$$\pi_h^t\left(a \mid s\right) \propto \exp\left(\frac{w_h^t\left(a \mid s\right)}{\lambda}\right), \tag{D.1}$$

for a vector $w_h^t(s) \in \mathbb{R}^{|\mathcal{A}|}$ satisfying $2\lambda H \log \sigma_{\min} \leq w_h^t\left(a \mid s\right) \leq H$. We prove it by induction on $t$. Suppose that there exist $t \in \mathbb{N}$ and $w_h^t$ satisfying (D.1). By the update rule (4.4), we have

$$\pi_h^{t+1}\left(a \mid s\right) \propto \left(\sigma_h\left(a \mid s\right)\right)^{\lambda\eta} \left(\pi_h^t\left(a \mid s\right)\right)^{1-\lambda\eta} \exp\left(\eta Q_h^{\lambda,\sigma}(s, a, \pi^t, \mu^t)\right)$$

$$\propto \exp\left(\frac{\lambda\eta w_h^\sigma\left(a \mid s\right) + (1-\eta\lambda)w_h^t\left(a \mid s\right) + \lambda\eta Q_h^{\lambda,\sigma}(s, a, \pi^t, \mu^t)}{\lambda}\right).$$

Set $w_h^{t+1}\left(a \mid s\right) := \lambda\eta w_h^\sigma\left(a \mid s\right) + (1-\eta\lambda)w_h^t(a|s) + \lambda\eta Q_h^{\lambda,\sigma}(s, a, \pi^t, \mu^t)$, we get $\pi_h^{t+1}(a|s) \propto e^{\frac{w_h^{t+1}(a \mid s)}{\lambda}}$. From Lemma E.3 and the hypothesis of the induction, we get $2\lambda H \log \sigma_{\min} \leq w_h^{t+1}\left(a \mid s\right) \leq H$.

Then we have for any $a_1, a_2 \in \mathcal{A}$:

$$\frac{\pi_h^t\left(a_1 \mid s\right)}{\pi_h^t\left(a_2 \mid s\right)} = \exp\left(\frac{w_h^t\left(a_1 \mid s\right) - w_h^t\left(a_2 \mid s\right)}{\lambda}\right) \leq \exp\left(\frac{H(1-\lambda\log\sigma_{\min})}{\lambda}\right).$$

It follows that:

$$\min_{a\in\mathcal{A}} \pi^t(a|s) \geq \exp\left(\frac{-H(1-\lambda\log\sigma_{\min})}{\lambda}\right) \max_{a'\in\mathcal{A}} \pi_h^t\left(a \mid s\right) \geq |\mathcal{A}|^{-1} \exp\left(\frac{-H(1-\lambda\log\sigma_{\min})}{\lambda}\right).$$

Therefore, we have:

$$\left\|\log\pi_h^t(s)\right\|_\infty \leq \frac{H(1-\lambda\log\sigma_{\min})}{\lambda} + \log|\mathcal{A}|.$$

From Lemmas E.1 and E.3, we have for $\pi_h^*$ and $a_1, a_2 \in \mathcal{A}$:

$$\frac{\pi_h^*\left(a_1 \mid s\right)}{\pi_h^*\left(a_2 \mid s\right)} = \exp\left(\frac{Q_h^{\lambda,\sigma}(s, a_1, \pi^t, \mu^t) + w_h^\sigma\left(a_1 \mid s\right) - Q_h^{\lambda,\sigma}(s, a_2, \pi^t, \mu^t) - w_h^\sigma\left(a_2 \mid s\right)}{\lambda}\right)$$

$$\leq \exp\left(\frac{H(1-\lambda\log\sigma_{\min})}{\lambda}\right),$$

and, we get $\|\log\pi_h^*(s)\|_\infty \leq \frac{H(1-\lambda\log\sigma_{\min})}{\lambda} + \log|\mathcal{A}|$. ∎

**Lemma D.3.** *Let* $G_h^{\lambda,\sigma}(s, a, \pi^t, \mu^t) := Q_h^{\lambda,\sigma}(s, a, \pi^t, \mu^t) - \lambda\log\dfrac{\pi_h^t\left(a \mid s\right)}{\sigma_h\left(a \mid s\right)}$.

$$\left|G_h^{\lambda,\sigma}(s, a, \pi^t, \mu^t) - G_h^{\lambda,\sigma}(s, a', \pi^t, \mu^t)\right|$$

$$\leq 2L\sum_{l=h}^{H}\left\|\mu_l^t - \mu_l^*\right\|_1 + C^{\lambda,\sigma,H,|\mathcal{A}|}\left(E_h(a, \pi^t, \varpi^*) + E_h(a', \pi^t, \varpi^*)\right),$$

*for* $a, a' \in \mathcal{A}$. *Here,*

$$C^{\lambda,\sigma,H,|\mathcal{A}|} := 2\lambda|\mathcal{A}|e^{\frac{H(1-\lambda\log\sigma_{\min})}{\lambda}} + 2(1+H) - \lambda(1+2H)\log\sigma_{\min} + 2\lambda\log|\mathcal{A}|,$$

*and*

$$E_h(a, \pi^t, \varpi^*) := \mathbb{E}\left[\sum_{l=h}^{H}\left\|\pi_l^*(s_l) - \pi_l^t(s_l)\right\|_1 \;\middle|\; \begin{array}{c} s_h = s, a_h = a, \\ s_{l+1} \sim P_l(s_l, a_l), \\ a_l \sim \varpi_l^*(s_l) \\ \text{for each } l \in \{h, \dots, H\} \end{array}\right].$$

**Proof of Lemma D.3.** We first compute the absolute value as follows:

$$\left| G_h^{\lambda,\sigma}(s,a,\pi^t,\mu^t) - G_h^{\lambda,\sigma}(s,a',\pi^t,\mu^t) \right|$$

$$= \left| \left( Q_h^{\lambda,\sigma}(s,a,\pi^t,\mu^t) - \lambda \log \frac{\pi_h^t(a\mid s)}{\sigma_h(a\mid s)} \right) - \left( Q_h^{\lambda,\sigma}(s,a',\pi^t,\mu^t) - \lambda \log \frac{\pi_h^t(a'\mid s)}{\sigma_h(a'\mid s)} \right) \right|$$

$$\leq \left| \left( Q_h^{\lambda,\sigma}(s,a,\varpi^*,\mu^*) - \lambda \log \frac{\pi_h^t(a\mid s)}{\sigma_h(a\mid s)} \right) - \left( Q_h^{\lambda,\sigma}(s,a',\varpi^*,\mu^*) - \lambda \log \frac{\pi_h^t(a'\mid s)}{\sigma_h(a'\mid s)} \right) \right|$$

$$+ \left| \left( Q_h^{\lambda,\sigma}(s,a,\pi^t,\mu^t) - Q_h^{\lambda,\sigma}(s,a,\varpi^*,\mu^*) \right) - \left( Q_h^{\lambda,\sigma}(s,a',\pi^t,\mu^t) - Q_h^{\lambda,\sigma}(s,a',\varpi^*,\mu^*) \right) \right|.$$

$$\tag{D.2}$$

By Lemmas D.2 and E.1, the first term of right-hand side in (D.3) can be computed as

$$\left| \left( Q_h^{\lambda,\sigma}(s,a,\varpi^*,\mu^*) - \lambda \log \frac{\pi_h^t(a\mid s)}{\sigma_h(a\mid s)} \right) - \left( Q_h^{\lambda,\sigma}(s,a',\varpi^*,\mu^*) - \lambda \log \frac{\pi_h^t(a'\mid s)}{\sigma_h(a'\mid s)} \right) \right|$$

$$= \left| \left( \lambda \log \frac{\varpi_h^*(a\mid s)}{\sigma_h(a\mid s)} - \lambda \log \frac{\pi_h^t(a\mid s)}{\sigma_h(a\mid s)} \right) - \left( \lambda \log \frac{\varpi_h^*(a'\mid s)}{\sigma_h(a'\mid s)} - \lambda \log \frac{\pi_h^t(a'\mid s)}{\sigma_h(a'\mid s)} \right) \right|$$

$$\leq \lambda \left( \left| \log \frac{\varpi_h^*(a\mid s)}{\pi_h^t(a\mid s)} \right| + \left| \log \frac{\varpi_h^*(a'\mid s)}{\pi_h^t(a'\mid s)} \right| \right)$$

$$\leq \lambda \left( \frac{1}{\varpi_{\min}^*} + \frac{1}{\min_{a\in\mathcal{A}} \pi_h^t(a\mid s)} \right) \left( \left| \varpi_h^*(a\mid s) - \pi_h^t(a\mid s) \right| + \left| \varpi_h^*(a'\mid s) - \pi_h^t(a'\mid s) \right| \right)$$

$$\leq 2\lambda|\mathcal{A}| \exp\left( \frac{H(1-\lambda\log\sigma_{\min})}{\lambda} \right) \left( \left| \varpi_h^*(a\mid s) - \pi_h^t(a\mid s) \right| + \left| \varpi_h^*(a'\mid s) - \pi_h^t(a'\mid s) \right| \right).$$

$$\tag{D.3}$$

By Proposition E.8 and Lemma E.6, the second term is bounded as

$$\left| \left( Q_h^{\lambda,\sigma}(s,a,\pi^t,\mu^t) - Q_h^{\lambda,\sigma}(s,a,\varpi^*,\mu^*) \right) - \left( Q_h^{\lambda,\sigma}(s,a',\pi^t,\mu^t) - Q_h^{\lambda,\sigma}(s,a',\varpi^*,\mu^*) \right) \right|$$

$$\leq 2L \sum_{l=h}^{H} \left\| \mu_l^t - \mu_l^* \right\|_1$$

$$+ C^{\lambda,\sigma}(\pi^t,\varpi^*) \, \mathbb{E}\left[ \sum_{l=h+1}^{H} \left\| \pi_l^*(s_l) - \pi_l^t(s_l) \right\|_1 \;\middle|\; \begin{matrix} s_{h+1} \sim P_h(\bullet\mid s,a), \\ s_{l+1} \sim P_l(s_l,a_l), \\ a_l \sim \varpi_l^*(s_l) \\ \text{for each } l \in \{h+1,\ldots,H\} \end{matrix} \right]$$

$$+ C^{\lambda,\sigma}(\pi^t,\varpi^*) \, \mathbb{E}\left[ \sum_{l=h+1}^{H} \left\| \pi_l^*(s_l) - \pi_l^t(s_l) \right\|_1 \;\middle|\; \begin{matrix} s_{h+1} \sim P_h(\bullet\mid s,a'), \\ s_{l+1} \sim P_l(s_l,a_l), \\ a_l \sim \varpi_l^*(s_l) \\ \text{for each } l \in \{h+1,\ldots,H\} \end{matrix} \right].$$

Furthermore, $C^{\lambda,\sigma}(\pi^t,\varpi^*)$ can be bounded as

$$C^{\lambda,\sigma}(\pi^t,\varpi^*) \leq 2 - \lambda\log\sigma_{\min} + 2\lambda\left( \frac{H(1-\lambda\log\sigma_{\min})}{\lambda} + \log|\mathcal{A}| \right)$$

$$= 2(1+H) - \lambda(1+2H)\log\sigma_{\min} + 2\lambda\log|\mathcal{A}|.$$

∎

**Proof of Theorem 4.3.** Set

$$C := 4H^2 \left( L^2H^2 + \frac{\left( C^{\lambda,\sigma,H,|\mathcal{A}|} \right)^2}{|\mathcal{A}| \exp\left( \frac{H(1-\lambda\log\sigma_{\min})}{\lambda} \right)} \right) \tag{D.4}$$

$$= 4H^2 \left( L^2H^2 + \frac{\left( 2\lambda|\mathcal{A}| e^{\frac{H(1-\lambda\log\sigma_{\min})}{\lambda}} + 2(1+H) - \lambda(1+2H)\log\sigma_{\min} + 2\lambda\log|\mathcal{A}| \right)^2}{|\mathcal{A}| e^{\frac{H(1-\lambda\log\sigma_{\min})}{\lambda}}} \right)$$

$$\eta^* = \min\left\{\frac{1}{2H\left(L + C^{\lambda,\sigma,H,|\mathcal{A}|}\right)}, \frac{\lambda}{2C}\right\}, \tag{D.5}$$

where $C^{\lambda,\sigma,H,|\mathcal{A}|}$ is the constant defined in Lemma D.3. We prove the inequality by induction on $t$.

**(I) Base step $t = 0$:** It is obvious.

**(II) Inductive step:** Suppose that there exists $t \in \mathbb{N}$ such that $\pi^t \in \Omega$. Lemma D.1 yields that

$$
\begin{aligned}
&D_{\mu^*}(\varpi^*, \pi^{t+1}) - D_{\mu^*}(\varpi^*, \pi^t) - D_{\mu^*}(\pi^t, \pi^{t+1}) \\
&= \sum_{h=1}^{H} \mathbb{E}_{s \sim \mu_h^*}\left[\left\langle \log \frac{\pi_h^t(s)}{\pi_h^{t+1}(s)}, (\varpi_h^* - \pi_h^t)(s) \right\rangle\right] \\
&= -\sum_{h=1}^{H} \mathbb{E}_{s \sim \mu_h^*}\left[\left\langle \frac{\eta}{1 - \lambda\eta}\left(Q_h^{\lambda,\sigma}(s, \bullet, \pi^t, \mu^t) - \lambda \log \frac{\pi_h^{t+1}(s)}{\sigma_h(s)}\right), (\varpi_h^* - \pi_h^t)(s) \right\rangle\right] \\
&= -\frac{\eta}{1 - \lambda\eta} \underbrace{\sum_{h=1}^{H} \mathbb{E}_{s \sim \mu_h^*}\left[\left\langle Q_h^{\lambda,\sigma}(s, \bullet, \pi^t, \mu^t), (\varpi_h^* - \pi_h^t)(s) \right\rangle\right]}_{=:I} \\
&\quad + \frac{\lambda\eta}{1 - \lambda\eta} \sum_{h=1}^{H} \mathbb{E}_{s \sim \mu_h^*}\left[\left\langle \log \frac{\pi_h^{t+1}(s)}{\sigma_h(s)}, (\varpi_h^* - \pi_h^{t+1})(s) \right\rangle\right] \\
&\leq -\frac{\eta}{1 - \lambda\eta}\left(\lambda D_{\mu^*}(\varpi^*, \sigma) - \lambda D_{\mu^*}(\pi^{t+1}, \sigma)\right) \\
&\quad + \frac{\lambda\eta}{1 - \lambda\eta}\left(D_{\mu^*}(\varpi^*, \sigma) - D_{\mu^*}(\varpi^*, \pi^{t+1}) - D_{\mu^*}(\pi^{t+1}, \sigma)\right) \\
&\leq -\frac{\lambda\eta}{1 - \lambda\eta} D_{\mu^*}(\varpi^*, \pi^{t+1}),
\end{aligned} \tag{D.6}
$$

where I is bounded from below as follows: By Lemma E.4, we get

$$I = J^{\lambda,\sigma}(\mu^{t+1}, \varpi^*) - J^{\lambda,\sigma}(\mu^{t+1}, \pi^{t+1}) + \lambda D_{\mu^*}(\varpi^*, \sigma) - \lambda D_{\mu^*}(\pi^{t+1}, \sigma). \tag{D.7}$$

By virtue of the definition of mean-field equilibrium and Lemma E.2, we find

$$J^{\lambda,\sigma}(\mu^{t+1}, \varpi^*) - J^{\lambda,\sigma}(\mu^{t+1}, \pi^{t+1}) \geq J^{\lambda,\sigma}(\mu^*, \varpi^*) - J^{\lambda,\sigma}(\mu^*, \pi^{t+1}) \geq 0.$$

Then, we obtain

$$I \geq \lambda D_{\mu^*}(\varpi^*, \sigma) - \lambda D_{\mu^*}(\pi^{t+1}, \sigma).$$

For the last term $D_{\mu^*}(\pi^t, \pi^{t+1})$ of the leftmost hand of (D.6), we can employ a similar argument to (Abe et al., 2023, Lemma 5.4), that is, we can estimate $D_{\mu^*}(\pi^t, \pi^{t+1})$ as follows: Set $G(a) :=$ $G_h^{\lambda,\sigma}(s, a, \pi^t, \mu^t) = Q_h^{\lambda,\sigma}(s, a, \pi^t, \mu^t) - \lambda \log \frac{\pi_h^t(a \mid s)}{\sigma_h(a \mid s)}$. Note that $\max_{a,a' \in \mathcal{A}} |G(a') - G(a)| \leq \eta^{*-1}$ by Lemma D.3. By the update rule (4.4) and concavity of the logarithmic function $\log$, we

have

$$D_{\mu^*}(\pi^t, \pi^{t+1})$$

$$= \sum_{h=1}^{H} \mathbb{E}_{s \sim \mu_h^*} \left[ \sum_{a \in \mathcal{A}} \pi_h^t(a \mid s) \log \frac{\pi_h^t(a \mid s)}{\pi_h^{t+1}(a \mid s)} \right]$$

$$= \sum_{h=1}^{H} \mathbb{E}_{s \sim \mu_h^*} \left[ \sum_{a \in \mathcal{A}} \pi_h^t(a \mid s) \log \frac{\sum\limits_{a' \in \mathcal{A}} (\sigma_h(a' \mid s))^{\lambda \eta} (\pi_h^t(a' \mid s))^{1-\lambda \eta} \exp\left(\eta Q_h^{\lambda,\sigma}(s, a', \pi^t, \mu^t)\right)}{(\sigma_h(a \mid s))^{\lambda \eta} (\pi_h^t(a \mid s))^{-\lambda \eta} \exp\left(\eta Q_h^{\lambda,\sigma}(s, a, \pi^t, \mu^t)\right)} \right]$$

$$= \sum_{h=1}^{H} \mathbb{E}_{s \sim \mu_h^*} \left[ \sum_{a \in \mathcal{A}} \pi_h^t(a \mid s) \log \frac{\sum\limits_{a' \in \mathcal{A}} \pi_h^t(a' \mid s) \exp\left(\eta Q_h^{\lambda,\sigma}(s, a', \pi^t, \mu^t) - \lambda \eta \log \frac{\pi_h^t(a' \mid s)}{\sigma_h(a' \mid s)}\right)}{\exp\left(\eta Q_h^{\lambda,\sigma}(s, a, \pi^t, \mu^t) - \lambda \eta \log \frac{\pi_h^t(a \mid s)}{\sigma_h(a \mid s)}\right)} \right]$$

$$\leq \sum_{h=1}^{H} \mathbb{E}_{s \sim \mu_h^*} \left[ \log \sum_{a \in \mathcal{A}} \pi_h^t(a \mid s) \frac{\sum\limits_{a' \in \mathcal{A}} \pi_h^t(a' \mid s) \exp\left(\eta Q_h^{\lambda,\sigma}(s, a', \pi^t, \mu^t) - \lambda \eta \log \frac{\pi_h^t(a' \mid s)}{\sigma_h(a' \mid s)}\right)}{\exp\left(\eta Q_h^{\lambda,\sigma}(s, a, \pi^t, \mu^t) - \lambda \eta \log \frac{\pi_h^t(a \mid s)}{\sigma_h(a \mid s)}\right)} \right].$$

$$\text{(D.8)}$$

If we take $\eta$ to be $\eta \leq \eta^*$, it follows that

$$\eta(G(a') - G(a)) \leq 1,$$

for $a, a' \in \mathcal{A}$. Thus, we can use the inequality $e^x \leq 1 + x + x^2$ for $x \leq 1$ and obtain

$$D_{\mu^*}(\pi^t, \pi^{t+1})$$

$$\leq \sum_{h=1}^{H} \mathbb{E}_{s \sim \mu_h^*} \left[ \log \sum_{a,a' \in \mathcal{A}} \pi_h^t(a \mid s) \pi_h^t(a' \mid s) e^{\eta(G(a') - G(a))} \right]$$

$$=$$

$$\leq \sum_{h=1}^{H} \mathbb{E}_{s \sim \mu_h^*} \left[ \log \sum_{a,a' \in \mathcal{A}} \pi_h^t(a \mid s) \pi_h^t(a' \mid s) \left(1 + \eta(G(a') - G(a)) + \eta^2(G(a') - G(a))^2\right) \right]$$

$$= \sum_{h=1}^{H} \mathbb{E}_{s \sim \mu_h^*} \left[ \log \sum_{a,a' \in \mathcal{A}} \pi_h^t(a \mid s) \pi_h^t(a' \mid s) \left(1 + (G(a') - G(a))^2\right) \right]$$

$$= \sum_{h=1}^{H} \mathbb{E}_{s \sim \mu_h^*} \left[ \log \left(1 + \eta^2 \sum_{a,a' \in \mathcal{A}} \pi_h^t(a \mid s) \pi_h^t(a' \mid s) (G(a') - G(a))^2\right) \right]$$

$$\leq \eta^2 \sum_{h=1}^{H} \mathbb{E}_{s \sim \mu_h^*} \left[ \sum_{a,a' \in \mathcal{A}} \pi_h^t(a \mid s) \pi_h^t(a' \mid s) (G(a') - G(a))^2 \right].$$

By [Lemma D.3](), we can see that

$$\sum_{a,a' \in \mathcal{A}} \pi_h^t(a \mid s) \pi_h^t(a' \mid s) (G(a') - G(a))^2$$

$$\leq \sum_{a,a' \in \mathcal{A}} \pi_h^t(a \mid s) \pi_h^t(a' \mid s) \left(2L \sum_{l=h}^{H} \left\|\mu_l^t - \mu_l^*\right\|_1 + C^{\lambda,\sigma,H,|\mathcal{A}|}\left(E_h(a, \pi^t, \varpi^*) + E_h(a', \pi^t, \varpi^*)\right)\right)^2$$

$$\leq \sum_{a,a' \in \mathcal{A}} \pi_h^t(a \mid s) \pi_h^t(a' \mid s) \left(8L^2\left(\sum_{l=h}^{H} \left\|\mu_l^t - \mu_l^*\right\|_1\right)^2 + 4\left(C^{\lambda,\sigma,H,|\mathcal{A}|}\right)^2\left(E_h^2(a, \pi^t, \varpi^*) + E_h^2(a', \pi^t, \varpi^*)\right)\right)$$

$$
\leq 8L^2 H \sum_{l=h}^{H} \left\| \mu_l^t - \mu_l^* \right\|_1^2 + 8 \left( C^{\lambda,\sigma,H,|\mathcal{A}|} \right)^2 \sum_{a \in \mathcal{A}} \pi_h^t \left( a \mid s \right) E_h^2(a, \pi^t, \varpi^*)
$$

$$
= 8L^2 H \sum_{l=h}^{H} \left\| \mu_l^t - \mu_l^* \right\|_1^2 + 8 \left( C^{\lambda,\sigma,H,|\mathcal{A}|} \right)^2 \sum_{a \in \mathcal{A}} \frac{\pi_h^t \left( a \mid s \right)}{\varpi_h^* \left( a \mid s \right)} \varpi_h^* \left( a \mid s \right) E_h^2(a, \pi^t, \varpi^*)
$$

$$
\leq 8L^2 H \sum_{l=h}^{H} \left\| \mu_l^t - \mu_l^* \right\|_1^2 + \frac{8 \left( C^{\lambda,\sigma,H,|\mathcal{A}|} \right)^2}{|\mathcal{A}| \exp \left( \frac{H(1-\lambda \log \sigma_{\min})}{\lambda} \right)} \sum_{a \in \mathcal{A}} \varpi_h^* \left( a \mid s \right) E_h^2(a, \pi^t, \varpi^*)
$$

$$
\leq 8L^2 H \sum_{l=h}^{H} \left\| \mu_l^t - \mu_l^* \right\|_1^2 + \frac{8H \left( C^{\lambda,\sigma,H,|\mathcal{A}|} \right)^2}{|\mathcal{A}| \exp \left( \frac{H(1-\lambda \log \sigma_{\min})}{\lambda} \right)} \sum_{l=h}^{H} \mathbb{E}_{s_l \sim \mu_l^*} \left[ \left\| \pi_l^*(s_l) - \pi_l^t(s_l) \right\|_1^2 \right]
$$

$$
\leq 8L^2 H \sum_{l=h}^{H} \left\| \mu_l^t - \mu_l^* \right\|_1^2 + \frac{4H \left( C^{\lambda,\sigma,H,|\mathcal{A}|} \right)^2}{|\mathcal{A}| \exp \left( \frac{H(1-\lambda \log \sigma_{\min})}{\lambda} \right)} D_{\mu^*}(\varpi^*, \pi^t).
$$

Moreover, [Lemma E.6](#) bounds $\sum_{l=h}^{H} \left\| \mu_l^t - \mu_l^* \right\|_1^2$ as

$$
\sum_{l=h}^{H} \left\| \mu_l^t - \mu_l^* \right\|_1^2
$$

$$
\leq \sum_{l=h}^{H} \left( \sum_{k=0}^{l-1} \mathbb{E}_{s_k \sim \mu_k^*} \left[ \left\| \pi_k^*(s_k) - \pi_k^t(s_k) \right\| \right] \right)^2
$$

$$
\leq H \sum_{l=h}^{H} \sum_{k=0}^{l-1} \mathbb{E}_{s_k \sim \mu_k^*} \left[ \left\| \pi_k^*(s_k) - \pi_k^t(s_k) \right\|^2 \right]
$$

$$
\leq \frac{1}{2} H^2 D_{\mu^*}(\varpi^*, \pi^t).
$$

Therefore, we finally obtain

$$
D_{\mu^*}(\varpi^*, \pi^{t+1}) \leq \left( 1 - \lambda \eta + C \eta^2 \right) D_{\mu^*}(\varpi^*, \pi^t) \leq \left( 1 - \frac{1}{2} \lambda \eta \right) D_{\mu^*}(\varpi^*, \pi^t), \tag{D.9}
$$

where we use $C\eta \leq C\eta^* \leq 1/2$. ∎

# E  USEFUL LEMMAS

For Mean-field games, one can write down the *Bellman optimality equation* as follows: for a function $Q' \colon \mathcal{S} \to \Delta(\mathcal{A})$, a policy $\pi' \colon \mathcal{S} \to \Delta(\mathcal{A})$, $\sigma' \colon \mathcal{S} \to \Delta(\mathcal{A})$ and $s \in \mathcal{S}$ set

$$
f_s^{\sigma'}(Q', \pi') = \langle Q'(s), \pi'(s) \rangle - \lambda D_{\mathrm{KL}}(\pi'(s), \sigma'(s)). \tag{E.1}
$$

**Lemma E.1.** *Let $(\mu^*, \varpi^*)$ be equilibrium in the sense of [Definition 4.2](#). Then, it holds that*

$$
\varpi_h^*(s) = \underset{p \in \Delta(\mathcal{A})}{\arg \max} \, f_s^{\sigma_h} \left( Q_h^{\lambda,\sigma}(s, \bullet, \varpi^*, \mu^*), p \right) \propto \sigma_h \left( \bullet \mid s \right) \exp \left( \frac{Q_h^{\lambda,\sigma}(s, \bullet, \varpi^*, \mu^*)}{\lambda} \right),
$$

*for each $s \in \mathcal{S}$ and $h \in [H]$. Moreover,*

$$
\left\langle Q_h^{\lambda,\sigma}(s, \bullet, \varpi^*, \mu^*) - \lambda \log \frac{\pi_h^*(s)}{\sigma_h(s)}, \delta \right\rangle = 0,
$$

*for all $\delta \in \mathbb{R}^{|\mathcal{A}|}$ such that $\sum_a \delta(a) = 0$.*

*Proof.* See the Bellman optimality equation (e.g., ([Agarwal et al., 2022](#), Theorem 1.9)). ∎

**Lemma E.2.** *Under [Assumption 2.2](#), it holds that, for all $\pi, \widetilde{\pi} \in (\Delta(\mathcal{A})^{\mathcal{S}})^H$,*

$$J^{\lambda,\sigma}(m[\pi], \pi) + J^{\lambda,\sigma}(m[\widetilde{\pi}], \widetilde{\pi}) - J^{\lambda,\sigma}(m[\pi], \widetilde{\pi}) - J^{\lambda,\sigma}(m[\widetilde{\pi}], \pi) \leq 0,$$

*where $m$ is defined in [(2.1)](#).*

***Proof of [Lemma E.2](#).*** The proof is similar to ([Zhang et al., 2023](#), §H). Set $\mu = m[\pi]$ and $\widetilde{\mu} = m[\widetilde{\pi}]$. One can obtain that

$$J^{\lambda,\sigma}(m[\pi], \pi) + J^{\lambda,\sigma}(m[\widetilde{\pi}], \widetilde{\pi}) - J^{\lambda,\sigma}(m[\pi], \widetilde{\pi}) - J^{\lambda,\sigma}(m[\widetilde{\pi}], \pi)$$

$$= (J^{\lambda,\sigma}(\mu, \pi) - J^{\lambda,\sigma}(\widetilde{\mu}, \pi)) + (J^{\lambda,\sigma}(\widetilde{\mu}, \widetilde{\pi}) - J^{\lambda,\sigma}(\mu, \widetilde{\pi}))$$

$$= \sum_{h=1}^{H} \sum_{s_h \in \mathcal{S}} m[\pi]_h(s_h) \sum_{a_h \in \mathcal{A}} \pi_h(a_h \mid s_h) (r_h(s_h, a_h, \mu_h) - r_h(s_h, a_h, \widetilde{\mu}_h))$$

$$+ \sum_{h=1}^{H} \sum_{s_h \in \mathcal{S}} m[\widetilde{\pi}]_h(s_h) \sum_{a_h \in \mathcal{A}} \widetilde{\pi}_h(a_h \mid s_h) (r_h(s_h, a_h, \widetilde{\mu}_h) - r_h(s_h, a_h, \mu_h))$$

$$= \sum_{h,s,a} (\pi_h(a \mid s) \mu_h(s) - \widetilde{\pi}_h(a \mid s) \widetilde{\mu}_h(s)) (r_h(s_h, a_h, \mu_h) - r_h(s_h, a_h, \widetilde{\mu}_h)),$$

and the right-hand side of the above inequality is less than 0 by [Assumption 2.2](#). ∎

**Lemma E.3.** *Let $V_h^{\lambda,\sigma}$ be the state value function defined in [(4.2)](#) and $Q_h^{\lambda,\sigma}$ be the state action value function defined in [(4.3)](#). For any $s \in \mathcal{A}$, $a \in \mathcal{A}$, and $h \in [H]$, it holds that*

$$\lambda(H - h + 1) \log \sigma_{\min} \leq V_h^{\lambda,\sigma}(s, \mu, \pi) \leq H - h + 1,$$

$$\lambda(H - h + 1) \log \sigma_{\min} \leq Q_h^{\lambda,\sigma}(s, a, \mu, \pi) \leq H - h + 2.$$

***Proof.*** We prove the inequalities by backward induction on $h$. By definition, we have

$$\max_{s \in \mathcal{S}} V_h^{\lambda,\sigma}(s, \mu, \pi) = \mathbb{E}\left[\sum_{l=h}^{H} (r_l(s_l, a_l, \mu_l) - \lambda D_{\mathrm{KL}}(\pi_l(s_l), \sigma_l(s_l))) \,\middle|\, s_h = s\right]$$

$$= \langle r_h(s, \bullet, \mu_h), \pi_h(s) \rangle - \lambda D_{\mathrm{KL}}(\pi_h(s_h), \sigma_h(s_h))$$

$$+ \sum_{s_{h+1} \in \mathcal{S}} V_{h+1}^{\lambda,\sigma}(s_{h+1}, \mu, \pi) \sum_{a_h \in \mathcal{A}} P_h(s_{h+1} \mid s, a_h) \pi_h(a_h \mid s)$$

$$\leq 1 + \max_{s_{h+1} \in \mathcal{S}} V_{h+1}^{\lambda,\sigma}(s_{h+1}, \mu, \pi),$$

and

$$\min_{s \in \mathcal{S}} V_h^{\lambda,\sigma}(s, \mu, \pi) = \langle r_h(s, \bullet, \mu_h), \pi_h(s) \rangle - \lambda D_{\mathrm{KL}}(\pi_h(s_h), \sigma_h(s_h))$$

$$+ \sum_{s_{h+1} \in \mathcal{S}} V_{h+1}^{\lambda,\sigma}(s_{h+1}, \mu, \pi) \sum_{a_h \in \mathcal{A}} P_h(s_{h+1} \mid s, a_h) \pi_h(a_h \mid s)$$

$$\geq \lambda \log \sigma_{\min} + \max_{s_{h+1} \in \mathcal{S}} V_{h+1}^{\lambda,\sigma}(s_{h+1}, \mu, \pi).$$

Then, we have

$$V_h^{\lambda,\sigma}(s, \mu, \pi) \in [\lambda(H - h + 1) \log \sigma_{\min}, H - h + 1],$$

by the induction. The definition of $Q_h^{\lambda,\sigma}$ in [(4.3)](#) immediately yields the bound. ∎

**Lemma E.4.** *For all $\pi, \widetilde{\pi} \in (\Delta(\mathcal{A})^{\mathcal{S}})^H$, it holds that*

$$\sum_{h=1}^{H} \mathbb{E}_{s \sim m[\widetilde{\pi}]_h} \left[ \left\langle (\pi_h - \widetilde{\pi}_h)(s), Q_h^{\lambda,\sigma}(s, \bullet, \pi, \mu) \right\rangle \right] = J^{\lambda,\sigma}(\mu, \pi) - J^{\lambda,\sigma}(\mu, \widetilde{\pi}) - \lambda D_{m[\widetilde{\pi}]}(\widetilde{\pi}, \sigma) + \lambda D_{m[\widetilde{\pi}]}(\pi, \sigma).$$

**Proof.** From the definition of $V^{\lambda,\sigma}$ and $Q^{\lambda,\sigma}$ in (4.2) and (4.3), we have

$$\sum_{h=1}^{H} \mathbb{E}_{s\sim m[\widetilde{\pi}]_h} \left[ \left\langle \pi_h(s), Q_h^{\lambda,\sigma}(s,\bullet,\pi,\mu) \right\rangle \right]$$

$$= \sum_{h=1}^{H} \mathbb{E}_{s\sim m[\widetilde{\pi}]_h} \left[ \left\langle \pi_h(s), r_h(s,\bullet,\mu_h) + \mathbb{E}\left[ V_{h+1}^{\lambda,\sigma}(s_{h+1},\mu,\pi) \;\middle|\; s_{h+1}\sim P(s,\bullet,\mu_h) \right] \right\rangle \right]$$

$$= \sum_{h=1}^{H} \mathbb{E}_{s_h\sim m[\widetilde{\pi}]_h} \left[ \mathbb{E}_{a_h\sim\pi_h(s)}\left[ r_h(s_h,a_h,\mu_h) - \lambda D_{\mathrm{KL}}(\pi(s_h),\sigma(s_h)) \right] \right] + \lambda D_{m[\widetilde{\pi}]}(\pi,\sigma)$$

$$+ \sum_{h=1}^{H} \mathbb{E}_{s\sim m[\widetilde{\pi}]_h} \left[ \mathbb{E}\left[ V_{h+1}^{\lambda,\sigma}(s_{h+1},\mu,\pi) \;\middle|\; s_{h+1}\sim P(s,a_h,\mu_h), a_h\sim\pi_h(s) \right] \right] \tag{E.2}$$

$$= \sum_{h=1}^{H} \mathbb{E}_{s_h\sim m[\widetilde{\pi}]_h} \left[ V_h^{\lambda,\sigma}(s_h,\mu,\pi) - \mathbb{E}\left[ V_{h+1}^{\lambda,\sigma}(s_{h+1},\mu,\pi) \;\middle|\; \begin{array}{c} s_{h+1}\sim P(s,a_h,\mu_h), \\ a_h\sim\pi_h(s) \end{array} \right] \right]$$

$$+ \lambda D_{m[\widetilde{\pi}]}(\pi,\sigma)$$

$$+ \sum_{h=1}^{H} \mathbb{E}_{s\sim m[\widetilde{\pi}]_h} \left[ \mathbb{E}\left[ V_{h+1}^{\lambda,\sigma}(s_{h+1},\mu,\pi) \;\middle|\; \begin{array}{c} s_{h+1}\sim P(s,a_h,\mu_h), \\ a_h\sim\pi_h(s) \end{array} \right] \right]$$

$$= \sum_{h=1}^{H} \mathbb{E}_{s\sim m[\widetilde{\pi}]_h} \left[ V_h^{\lambda,\sigma}(s,\mu,\pi) \right] + \lambda D_{m[\widetilde{\pi}]}(\pi,\sigma).$$

Similarly, (4.1) and (2.1) gives us

$$\sum_{h=1}^{H} \mathbb{E}_{s\sim m[\widetilde{\pi}]_h} \left[ \left\langle \widetilde{\pi}_h(s), Q_h^{\lambda,\sigma}(s,\bullet,\pi,\mu) \right\rangle \right]$$

$$= \sum_{h=1}^{H} \mathbb{E}_{s_h\sim m[\widetilde{\pi}]_h} \left[ \mathbb{E}_{a_h\sim\widetilde{\pi}_h(s)}\left[ r_h(s_h,a_h,\mu_h) - \lambda D_{\mathrm{KL}}(\widetilde{\pi}(s_h),\sigma(s_h)) \right] \right] + \lambda D_{m[\widetilde{\pi}]}(\widetilde{\pi},\sigma)$$

$$+ \sum_{h=1}^{H} \mathbb{E}_{s\sim m[\widetilde{\pi}]_h} \left[ \mathbb{E}\left[ V_{h+1}^{\lambda,\sigma}(s_{h+1},\mu,\pi) \;\middle|\; s_{h+1}\sim P(s,a_h,\mu_h), a_h\sim\widetilde{\pi}_h(s) \right] \right] \tag{E.3}$$

$$= J^{\lambda,\sigma}(\mu,\widetilde{\pi}) + \lambda D_{m[\widetilde{\pi}]}(\widetilde{\pi},\sigma) + \sum_{h=1}^{H} \mathbb{E}_{s\sim m[\widetilde{\pi}]_{h+1}} \left[ V_{h+1}^{\lambda,\sigma}(s,\mu,\pi) \right].$$

Combining (E.2) and (E.3) yields

$$\sum_{h=1}^{H} \mathbb{E}_{s\sim m[\widetilde{\mu}]_h} \left[ \left\langle (\pi_h - \widetilde{\pi}_h)(s), Q_h^{\lambda,\sigma}(s,\bullet,\pi,\mu) \right\rangle \right]$$

$$= \left( \sum_{h=1}^{H} \mathbb{E}_{s\sim m[\widetilde{\pi}]_h} \left[ V_h^{\lambda,\sigma}(s,\mu,\pi) \right] + \lambda D_{m[\widetilde{\pi}]}(\pi,\sigma) \right)$$

$$- \left( J^{\lambda,\sigma}(\mu,\widetilde{\pi}) + \lambda D_{m[\widetilde{\pi}]}(\widetilde{\pi},\sigma) + \sum_{h=1}^{H} \mathbb{E}_{s\sim m[\widetilde{\pi}]_{h+1}} \left[ V_{h+1}^{\lambda,\sigma}(s,\mu,\pi) \right] \right)$$

$$= \left( \mathbb{E}_{s\sim m[\widetilde{\pi}]_1} \left[ V_1^{\lambda,\sigma}(s,\mu,\pi) \right] + \lambda D_{m[\widetilde{\pi}]}(\pi,\sigma) \right) - \left( J^{\lambda,\sigma}(\mu,\widetilde{\pi}) + \lambda D_{m[\widetilde{\pi}]}(\widetilde{\pi},\sigma) \right)$$

$$= \mathbb{E}_{s\sim\mu_1} \left[ V_1^{\lambda,\sigma}(s,\mu,\pi) \right] - J^{\lambda,\sigma}(\mu,\widetilde{\pi}) + \lambda D_{m[\widetilde{\pi}]}(\pi,\sigma) - \lambda D_{m[\widetilde{\pi}]}(\widetilde{\pi},\sigma),$$

which concludes the proof. ∎

**Lemma E.5.** *For all* $\pi, \widetilde{\pi} \in (\Delta(\mathcal{A})^{\mathcal{S}})^H$, *it holds that*

$$\sum_{h=1}^{H} \mathbb{E}_{s\sim m[\widetilde{\pi}]_h} \left[ \left\langle (\pi_h - \widetilde{\pi}_h)(s), \log\frac{\pi_h(s)}{\sigma_h(s)} \right\rangle \right] = D_{m[\widetilde{\pi}]}(\pi,\sigma) - D_{m[\widetilde{\pi}]}(\widetilde{\pi},\sigma) + D_{\widetilde{\pi}}(\widetilde{\pi},\pi).$$

**Proof.** A direct computation yields

$$\sum_{h=1}^{H} \mathbb{E}_{s \sim m[\widetilde{\pi}]_h} \left[ \left\langle (\pi_h - \widetilde{\pi}_h)(s), \log \frac{\pi_h(s)}{\sigma_h(s)} \right\rangle \right]$$

$$= D_{m[\widetilde{\pi}]}(\pi, \sigma) - \sum_{h=1}^{H} \mathbb{E}_{s \sim m[\widetilde{\pi}]_h} \left[ \left\langle \widetilde{\pi}_h(s), \log \frac{\widetilde{\pi}_h(s)}{\sigma_h(s)} - \log \frac{\widetilde{\pi}(s)}{\pi(s)} \right\rangle \right]$$

$$= D_{m[\widetilde{\pi}]}(\pi, \sigma) - D_{m[\widetilde{\pi}]}(\widetilde{\pi}, \sigma) + D_{m[\widetilde{\pi}]}(\widetilde{\pi}, \pi). \qquad \blacksquare$$

**Lemma E.6.** *The operator $m$ defined in* (2.1) *is 1-Lipschitz, namely, it holds that*

$$\|m[\pi]_{h+1} - m[\pi']_{h+1}\| \leq \sum_{l=0}^{h} \mathbb{E}_{s_l \sim m[\pi]_l} [\|\pi_l(s_l) - \pi'_l(s_l)\|], \qquad (E.4)$$

*for $\pi, \pi' \in (\Delta(\mathcal{A})^{\mathcal{S}})^H$ and all $h \in \{0, \ldots, H\}$. Here, we set $\pi_0(s) = \pi'_0(s) = U_{\mathcal{A}}$ for all $s \in \mathcal{S}$.*

**Proof.** Fix $\pi, \pi' \in (\Delta(\mathcal{A})^{\mathcal{S}})^H$. We prove the inequality by induction on $h$.

**(I) Base step $h = 0$:** It is obvious because $\|m[\pi]_1 - m[\pi']_1\| = \|\mu_1 - \mu_1\| = 0$.

**(II) Inductive step:** Suppose that there exists $h \in [H]$ satisfying the inequality (E.4). By (2.1), we obtain

$$\|m[\pi]_{h+2} - m[\pi']_{h+2}\|$$

$$\leq \sum_{\substack{s_{h+2} \in \mathcal{S}, \\ (s_{h+1}, a_{h+1}) \in \mathcal{S} \times \mathcal{A}}} P_{h+1}(s_{h+2} \mid s_{h+1}, a_{h+1}) m[\pi]_{h+1}(s_{h+1}) |\pi_{h+1}(a_{h+1} \mid s_{h+1}) - \pi'_{h+1}(a_{h+1} \mid s_{h+1})|$$

$$+ \sum_{\substack{s_{h+2} \in \mathcal{S}, \\ (s_{h+1}, a_{h+1}) \in \mathcal{S} \times \mathcal{A}}} P_{h+1}(s_{h+2} \mid s_{h+1}, a_{h+1}) \pi'_{h+1}(a_{h+1} \mid s_{h+1}) |m[\pi]_{h+1}(s_{h+1}) - m[\pi']_{h+1}(s_{h+1})|$$

$$\leq \sum_{(s_{h+1}, a_{h+1}) \in \mathcal{S} \times \mathcal{A}} m[\pi]_{h+1}(s_{h+1}) |\pi_{h+1}(a_{h+1} \mid s_{h+1}) - \pi'_{h+1}(a_{h+1} \mid s_{h+1})|$$

$$+ \sum_{s_{h+1} \in \mathcal{S}} |m[\pi]_{h+1}(s_{h+1}) - m[\pi']_{h+1}(s_{h+1})|$$

$$= \mathbb{E}_{s_{h+1} \sim m[\pi]_{h+1}} \left[ \|\pi_{h+1}(s_{h+1}) - \pi'_{h+1}(s_{h+1})\| \right] + \|m[\pi]_{h+1} - m[\pi']_{h+1}\|.$$

By the hypothesis of the induction, we finally obtain

$$\|m[\pi]_{h+2} - m[\pi']_{h+2}\|$$

$$\leq \mathbb{E}_{s \sim m[\pi]_{h+1}} \left[ \|\pi_{h+1}(s) - \pi'_{h+1}(s)\| \right] + \sum_{l=1}^{h} \mathbb{E}_{s \sim m[\pi]_l} \|\pi_l(s) - \pi'_l(s)\|$$

$$\leq \sum_{l=1}^{h+1} \mathbb{E}_{s \sim m[\pi]_l} \|\pi_l(s) - \pi'_l(s)\|. \qquad \blacksquare$$

**Lemma E.7.** *Let $\pi, \pi' \in (\Delta(\mathcal{A})^{\mathcal{S}})^H$, $\mu, \mu' \in \Delta(\mathcal{S})^H$, $s \in \mathcal{S}$, and $h \in \{1, \ldots, H+1\}$. Assume*

$$\min_{(h,a,s) \in [H] \times \mathcal{A} \times \mathcal{S}} \min\{\pi_h(a \mid s), \pi'_h(a \mid s)\} > 0,$$

*and set $\mu_{H+1} = \mu'_{H+1} = U_{\mathcal{S}}$, $\pi_{H+1}(s) = \pi'_{H+1}(s) = U_{\mathcal{A}}$ for all $s \in \mathcal{S}$.*

$$\left| V_h^{\lambda,\sigma}(s, \pi, \mu) - V_h^{\lambda,\sigma}(s, \pi', \mu') \right|$$

$$\leq \mathbb{E} \left[ \sum_{l=h}^{H+1} \left( C^{\lambda,\sigma}(\pi, \pi') \|\pi_l(s_l) - \pi'_l(s_l)\|_1 + L \|\mu_l - \mu'_l\|_1 \right) \middle| \begin{array}{c} s_h = s, \\ s_{l+1} \sim P_l(s_l, a_l), \\ a_l \sim \pi_l(s_l) \\ \text{for each } l \in \{h, \ldots, H+1\} \end{array} \right]$$

*for Here, $C^{\lambda,\sigma}(\pi,\pi') > 0$ is defined in Proposition E.8, and the discrete time stochastic process $(s_l)_{l=h}^H$ is induced recursively as $s_{l+1} \sim P_l(s_l, a_l), a_l \sim \pi_l(s_l)$ for each $l \in \{h, \ldots, H-1\}$.*

**Proof.** Fix $\pi, \pi', \mu$ and $\mu'$. We prove the inequality by backward induction on $h$.

**(I) Base step $h = H+1$:** It is obvious because $\left| V_{H+1}^{\lambda,\sigma}(s,\pi,\mu) - V_{H+1}^{\lambda,\sigma}(s,\pi',\mu') \right| = |0 - 0| = 0$.

**(II) Inductive step:** Suppose that there exists $h \in [H]$ satisfying

$$
\left| V_{h+1}^{\lambda,\sigma}(s,\pi,\mu) - V_{h+1}^{\lambda,\sigma}(s,\pi',\mu') \right|
$$
$$
\leq \mathbb{E} \left[ \sum_{l=h+1}^{H+1} \left( C^{\lambda,\sigma}(\pi,\pi') \|\pi_l(s_l) - \pi_l'(s_l)\|_1 + L\|\mu_h - \mu_h'\|_1 \right) \middle| \begin{array}{c} s_{h+1} = s, \\ s_{l+1} \sim P_l(s_l, a_l), \\ a_l \sim \pi_l(s_l) \\ \text{for each } l \in \{h+1, \ldots, H+1\} \end{array} \right],
$$
(E.5)

for all $s \in \mathcal{S}$. By the definition of the value function in (4.2) and Assumption 2.3, we have

$$
\left| V_h^{\lambda,\sigma}(s,\pi,\mu) - V_h^{\lambda,\sigma}(s,\pi',\mu') \right|
$$
$$
\leq \left| \sum_{a_h \in \mathcal{A}} \left( \pi_h(a_h \mid s) r_h(s, a_h, \mu_h) - \pi_h'(a_h \mid s) r_h(s, a_h, \mu_h') \right) \right|
$$
$$
+ \lambda |D_{\mathrm{KL}}(\pi_h(s), \sigma_h(s)) - D_{\mathrm{KL}}(\pi_h'(s), \sigma_h(s))|
$$
$$
+ \left| \sum_{\substack{a_h \in \mathcal{A}, \\ s_{h+1} \in \mathcal{S}}} P_h(s_{h+1} \mid s, a_h) \left( \pi_h(a_h \mid s) V_{h+1}^{\lambda,\sigma}(s_{h+1}, \pi, \mu) - \pi'(a_h \mid s) V_{h+1}^{\lambda,\sigma}(s_{h+1}, \pi', \mu') \right) \right|
$$
$$
\leq \|\pi_h(s) - \pi_h'(s)\|_1 + \sum_{a_h \in \mathcal{A}} \pi_h(a_h \mid s) |r_h(s, a_h, \mu_h) - r_h(s, a_h, \mu_h')|
$$
$$
+ \lambda \left| \sum_{a_h \in \mathcal{A}} \left( \pi_h(a_h \mid s) \left( \log \frac{\pi_h(a_h \mid s)}{\sigma_h(a_h \mid s)} - 1 \right) - \pi_h'(a_h \mid s) \left( \log \frac{\pi_h'(a_h \mid s)}{\sigma_h(a_h \mid s)} - 1 \right) \right) \right|
$$
$$
+ \|\pi_h(s) - \pi_h'(s)\|_1
$$
$$
+ \sum_{\substack{a_h \in \mathcal{A}, \\ s_{h+1} \in \mathcal{S}}} P_h(s_{h+1} \mid s, a_h) \pi_h(a_h \mid s) \left| V_{h+1}^{\lambda,\sigma}(s_{h+1}, \pi, \mu) - V_{h+1}^{\lambda,\sigma}(s_{h+1}, \pi', \mu') \right|
$$
$$
\leq 2\|\pi_h(s) - \pi_h'(s)\|_1 + L\|\mu_h - \mu_h'\|_1
$$
$$
+ \lambda \max_{(h,a,s)} \log \frac{1}{(\sigma\pi\pi')_h(a \mid s)} \|\pi_h(s) - \pi_h'(s)\|_1
$$
$$
+ \sum_{\substack{a_h \in \mathcal{A}, \\ s_{h+1} \in \mathcal{S}}} P_h(s_{h+1} \mid s, a_h) \pi_h(a_h \mid s) \left| V_{h+1}^{\lambda,\sigma}(s_{h+1}, \pi, \mu) - V_{h+1}^{\lambda,\sigma}(s_{h+1}, \pi', \mu') \right|
$$
$$
\leq C^{\lambda,\sigma}(\pi,\pi') \|\pi_h(s) - \pi_h'(s)\|_1 + L\|\mu_h - \mu_h'\|_1
$$
$$
+ \mathbb{E} \left[ \left| V_{h+1}^{\lambda,\sigma}(s_{h+1}, \pi, \mu) - V_{h+1}^{\lambda,\sigma}(s_{h+1}, \pi', \mu') \right| \middle| \begin{array}{c} s_h = s, \\ s_{h+1} \sim P_h(s_h, a_h), \\ a_h \sim \pi_h(s_h) \end{array} \right].
$$

Combining the above inequality and the hypothesis of the induction completes the proof. ∎

**Proposition E.8.** *Let $Q^{\lambda,\sigma}$ be the function defined by (4.3), and $(\pi, \pi') \in \left( (\Delta(\mathcal{A})^{\mathcal{S}})^H \right)^2$ be policies with full supports. Under Assumptions 2.3 and 4.1, it holds that*

$$
\left| Q_h^{\lambda,\sigma}(s, a, \pi, \mu) - Q_h^{\lambda,\sigma}(s, a, \pi', \mu') \right|
$$

$$
\leq L \sum_{l=h}^{H} \| \mu_l - \mu'_l \| + C^{\lambda,\sigma}(\pi, \pi') \, \mathbb{E}_{(s_l)_{l=h+1}^{H}} \left[ \sum_{l=h+1}^{H} \| \pi_l(s_l) - \pi'_l(s_l) \| \ \middle| \ s_h = s \right],
$$

*for $(h, s, a) \in [H] \times \mathcal{S} \times \mathcal{A}$ and $\mu, \mu' \in \Delta(\mathcal{S})^H$. Here, the random variables $(s_l)_{l=h+1}^{H}$ follows the stochastic process starting from state $s$ at time $h$, induced from $P$ and $\pi$, and the function $C^{\lambda,\sigma} \colon \left( (\Delta(\mathcal{A})^{\mathcal{S}})^H \right)^2 \to \mathbb{R}$ is given by $C^{\lambda,\sigma}(\pi, \pi') = 2 - \lambda \inf_{(h,s,a) \in [H] \times \mathcal{S} \times \mathcal{A}} \log (\sigma \pi \pi')_h (a \mid s)$.*

***Proof of Proposition E.8.*** Let $h$ be larger than 2. By the definition of $Q_h^{\lambda,\sigma}$ given in (4.3) and Lemma E.7, we have

$$
\left| Q_{h-1}^{\lambda,\sigma}(s, a, \pi, \mu) - Q_{h-1}^{\lambda,\sigma}(s, a, \pi', \mu') \right|
$$

$$
\leq \left| r_{h-1}(s, a, \mu_{h-1}) - r_{h-1}(s, a, \mu'_{h-1}) \right| + \mathbb{E}_{s_h \sim P_{h-1}(s,a)} \left[ \left| V_h^{\lambda,\sigma}(s_h, \pi, \mu) - V_h^{\lambda,\sigma}(s_h, \pi', \mu') \right| \right]
$$

$$
\leq L \left\| \mu_{h-1} - \mu'_{h-1} \right\| + \mathbb{E}_{s_h \sim P_{h-1}(s,a)} \left[ \left| V_h^{\lambda,\sigma}(s_h, \pi, \mu) - V_h^{\lambda,\sigma}(s_h, \pi', \mu') \right| \right].
$$

Combining the above inequality and Lemma E.7 completes the proof. ∎

# F EXPERIMENT DETAILS

We ran experiments on a laptop with an 11th Gen Intel Core i7-1165G7 8-core CPU, 16GB RAM, running Windows 11 Pro with WSL. As is clear from Algorithm 2, our proposed method is deterministic. Thus, we ran the algorithm only once for each experimental setting. We implemented our proposed method using Python. The computation of $Q^{\lambda,\sigma}$ and $\mu$ in Algorithm 2 was based on the implementation provided by Fabian et al. (2023).

We show further details for Beach Bar Process. We set $H = 10, |\mathcal{S}| = 10, \mathcal{A} = \{-1, \pm 0, +1\}, \lambda = 0.1, \eta = 0.1$, and

$$
P_h \left( s' \mid s, a \right) = \begin{cases} 1 - \varepsilon & \text{if } a = \pm 0 \ \& \ s' = s, \\ \dfrac{\varepsilon}{2} & \text{if } a = \pm 1 \ \& \ s' = s \pm 1, \\ 0 & \text{otherwise,} \end{cases}
$$

where we choose $\epsilon = 0.1$. In addition, we initialize $\sigma^0$ and $\pi^0$ in Algorithm 2 as the uniform distributions on $\mathcal{A}$.

