# OpenReview forum: "Last Iterate Convergence in Monotone Mean Field Games"
_ICLR.cc/2025/Conference — Submitted to ICLR 2025_

### Official Review · Reviewer_b31R · 2024-11-02

**Soundness:** 3
**Presentation:** 2
**Contribution:** 2
**Rating:** 5
**Confidence:** 3

**Summary:**

The paper studied regularized MFG and proposed a mirror descent algorithm that can converge in $O(\log(1/\epsilon))$ iterations.

**Strengths:**

The paper is well-written and the proof is clear. The result is the first last-iterate convergence in regularized MFG to my best knowledge.

**Weaknesses:**

My main concern is that, with the monotonicity condition and the regularization $\lambda$, the results can be straightforwardly obtained given the existing proof of exponentially fast convergence of MD in monotone games. See [1] for details.

Specially, the relationship between $D_{\mu^\ast} (\pi^\ast, \pi^{t+1})$ and $D_{\mu^\ast} (\pi^\ast, \pi^{t})$ has already been derived in Zhang et al 2023. To obtain the exponentially fast convergence rate, you would only need to get the $1-\lambda \eta$ factor out of $D_{\mu^\ast} (\pi^\ast, \pi^{t})$. One common way to achieve this is to use the monotonicity condition. From what I understand, this paper uses the same technique that has been used in papers such as [1].


[1] Cen, Shicong, Yuting Wei, and Yuejie Chi. "Fast policy extragradient methods for competitive games with entropy regularization." Advances in Neural Information Processing Systems 34 (2021): 27952-27964.

**Questions:**

Can you elaborate on the technical contributions of this work? Specifically, why wouldn't a straightforward application of the analysis proposed in [1] work in regularized monotone mean field games?

---

> ### Author Response · Authors · 2024-11-20
> **Rebuttal by authors**
>
> Thank you for your detailed feedback and questions.
> First of all, we would like to emphasize again that, in addition to the convergence results for regularized MFG in $O(\log (1/\varepsilon))$ you mentioned (Theorem 4.3), our contributions also include the convergence of the proximal point (Algorithm 1, Theorem 3.1) and the proposal of a tractable algorithm (Algorithm 2).
>
> We would like to address your concerns and elaborate on the technical contributions of our work.
>
> Our result on the exponential convergence of regularized mirror descent is not merely an application of the techniques developed in the literature, such as [1].
> As you correctly point out, Zhang et al.derive the relationship between $ D_{\mu^\ast}(\pi^\ast, \pi^{t+1}) $ and $ D_{\mu^\ast}(\pi^\ast, \pi^t) $ with some discretization error.
> Despite this, we would like to emphasize that their discretization error is completely different from ours in Equation (4.6).
>
> Furthermore, obtaining the $(1 - \lambda \eta)$ factor from this relationship is non-trivial. The reason lies in the complexity of the Q function $ Q^{\lambda,\sigma}_h(s,a,\pi,\mu) $ in MFG, where $\mu$ is given by $\mu = m[\pi]$, making it more complex with respect to $\pi$.
> To explain in more detail by comparing the zero-sum Markov game focused in [1]: In zero-sum Markov games, the value of the Q function at time $ h $ is determined by the policy from __future__ times $ h+1 $ to $ H $.
> In contrast, in MFG, the Q function at time $ h $ also depends on the state distribution $\mu_h$, where $\mu_h = m[\pi]_h$ is determined by the policy from __past__ times $ 1 $ to $ h-1 $.
> As a result, the Q function depends highly non-linearly on the policy over the entire horizon, complicating the analysis of discretization errors.
>
> To address this difficulty, we made two key contributions:
>     - We derived a different discretization error $ D_{\mu^\ast}(\pi^t, \pi^{t+1}) $ from Zhang et al., as shown in Equation (4.6). This refined expression of the error facilitates clearer subsequent analysis.
>     - We discovered that our discretization error can be bounded by $ D_{\mu^\ast}(\pi^\ast, \pi^t) $. The key insight here is the claim stated in l.369-372: “over the sequence $ (\pi^t)_t $, the value function $ Q_h^{\lambda,\sigma} $ behaves well, almost as if it were a Lipschitz continuous function.”
>
> We hope this clarifies the technical contributions of our work and addresses your concerns. Thank you again for your valuable feedback.

---

> > ### Comment · Reviewer_b31R · 2024-11-22
> >
> > Thank you for your clarification.
> >
> > Please correct me if I am wrong. But the new discretization error seems to be a variant of Lemma 6 of Zhan et al, and with the $D_{\mu^\ast}(\pi^t, \pi^{t+1})$ etc defined on mean field game. Moreover, though the derivation is different from Zhang et al, the error $D_{\mu^\ast}(\pi^t, \pi^{t+1})$ is just the drift of the policy update, which is naturally controlled by exponential weight. To my understanding, the analysis of this term (page 19) is essentially just a variant of the classic exponential weight analysis. Could you highlight the technical contributions for the analysis?
> >
> > Zhan, W., Cen, S., Huang, B., Chen, Y., Lee, J. D., & Chi, Y. (2023). Policy mirror descent for regularized reinforcement learning: A generalized framework with linear convergence. SIAM Journal on Optimization, 33(2), 1061-1091.

---

> > > ### Author Response · Authors · 2024-11-24
> > >
> > > Thank you for your detailed comments and for the variant of Lemma 6 of Zhan et al.
> > > Our work is not merely a variant of existing papers, for the reasons explained in the following key points that distinguish our contributions:
> > >
> > > - **The difficulty of applying the Three-Point Lemma to MFGs**: The Three-Point Lemma cannot be directly applied to Mean Field Games (MFGs). The main reason is that the inner product $\langle Q^{k}(s),\pi^{k+1}(s)-p\rangle$ in the right-hand side of the three-point lemma concerns the policy at iteration index $k+1$, not $k$. In our analysis (as shown on page 18), this term is transformed into $\langle Q^{k}(s),\pi^{k}(s)-p \rangle$, which allows us to apply a crucial lemma (Lemma E.4) that holds for MFGs. This transformation is non-trivial and essential for our analysis. See the fourth line of Eq. (D.6) for more details.
> > >
> > > - **Discretization error analysis**: In the three-point lemma, the term $D_{h_s}(\pi^{(k+1)}, \pi^{(k)})$ appears as a discretization error. In contrast, our analysis derives a reverse version $D_{\mu^\ast}(\pi^k, \pi^{k+1})$. This distinction is significant, especially for non-symmetric divergences such as the KL divergence. The reverse order in our analysis is crucial for the theoretical guarantees we provide.
> > >
> > > - **Difficulty introduced by the iteration index $k$ in theoretical analysis**: The reviewer may perceive the shift in the iteration index $k$ as a minor detail. However, even such small shifts can have a significant impact on the theoretical analysis. For example, consider the differences of $k$ in the analysis methods of the proximal point method and the mirror descent method. These differences are not trivial and require careful handling to ensure rigorous theoretical results.

---

> > > > ### Author Response · Authors · 2024-11-28
> > > >
> > > > Thank you for your interest in the technical details of our paper. We have commented on your concerns. We have also revised the manuscript and strengthened the comparison with previous research: please also see the global comment. After reading these comments, we hope you will acknowledge our technical contribution and reconsider your rating.

---

### Official Review · Reviewer_AgLM · 2024-11-03

**Soundness:** 3
**Presentation:** 1
**Contribution:** 2
**Rating:** 5
**Confidence:** 4

**Summary:**

This paper studies the convergence performance in Monotone Mean Field Games. The major issue is that the focus of this paper are all well studied. Authors cannot identify the research gap and contribution in a comparison to existing study.

**Strengths:**

The paper is well structured and organized.

**Weaknesses:**

1. Position: Last-iterate convergence and learning regularized MFGs have been extensively studied. However, the discussion of related work in this manuscript is somewhat superficial. The authors should conduct a thorough literature review and clearly position their work within the existing literature. This will help avoid giving the audience the impression that their contribution is marginal, merely by applying a proximal point iteration to MFGs.

2. Exponential convergence of mirror descent: Using mirror descent to learn regularized mean field games (MFGs) is not a novel approach. To the best of my knowledge, previous studies have established a standard convergence rate of $T^{-1/2}$. In contrast, this work claims an exponential convergence rate of $e^{-T}$, which represents a significant improvement. The authors should clearly explain the specific challenges they addressed to achieve this enhanced convergence rate that previous work did not tackle. Additionally, providing numerical demonstrations of such exponential convergence is essential, given the magnitude of this improvement.

3. Presentation: the presentation of the paper is not yet suitable for scientific publication. Please avoid excessive use of colors, shapes, and unconventional layouts. Figure 1 lacks informative content. Additionally, some language and structural transitions are too casual and do not provide sufficient discussion to support the arguments presented convincingly.

4. The numerical experiments are limited. More results regarding the algorithm performance in a comparison to baselines (OMD, FP, Multi-scale, double loop, single loop) for MFGs should be provided.

**Questions:**

Please see details in weakness.

---

> ### Author Response · Authors · 2024-11-20
> **Rebuttal by authors (Part 1)**
>
> Thank you for reviewing our paper.
> We will now address your concerns about our contribution compared to existing research.
>
> >* Summary: This paper studies the convergence performance in Monotone Mean Field Games. The major issue is that the focus of this paper are all well studied. Authors cannot identify the research gap and contribution in a comparison to existing study.
> >* Weaknesses: Position: Last-iterate convergence and learning regularized MFGs have been extensively studied.
>
> We respectfully disagree with this argument.
> To the best of our knowledge, the only study that provides a last-iterate convergence guarantee is the one by Perolat et al. (2022).
> We would greatly appreciate it if you could provide specific references to other studies that have focused on the last-iterate convergence guarantee if you are aware of any.
>
>
> >* Weaknesses: However, the discussion of related work in this manuscript is somewhat superficial. The authors should conduct a thorough literature review and clearly position their work within the existing literature.
>
> While it might have been beneficial to mention existing results on last-iterate convergence for games beyond MFGs as part of the related work, we believe that the literature we discussed in the manuscript comprehensively covers the existing research on "last-iterate convergence and learning regularized MFGs" to the best of our knowledge.
> In fact, even the result on time-averaged convergence (which is different from last-iterate convergence) of mirror descent for regularized MFGs was only recently established by Zhang et al. in NeurIPS 2023.
>
> > This will help avoid giving the audience the impression that their contribution is marginal, merely by applying a proximal point iteration to MFGs.
>
> To the best of our knowledge, no previous research has applied proximal point (PP) iteration to MFGs.
> Applying PP to MFGs and performing the theoretical analysis is not just a straightforward application of known techniques from the literature.
> The dependence of the state distribution $\mu$ on the policy $\pi$ introduces unique challenges specific to MFGs, as discussed in l.180-182 and further elaborated in our responses.

---

> ### Author Response · Authors · 2024-11-20
> **Rebuttal by authors (Part 2)**
>
> >* Exponential convergence of mirror descent: Using mirror descent to learn regularized mean field games (MFGs) is not a novel approach. To the best of my knowledge, previous studies have established a standard convergence rate of $T^{−1/2}$. In contrast, this work claims an exponential convergence rate of $e^{−T}$, which represents a significant improvement. The authors should clearly explain the specific challenges they addressed to achieve this enhanced convergence rate that previous work did not tackle.
>
> First, we would like to differentiate our results from the prior work of [Zhang et al. 2023], again, to make it clear.
> Our improvement is not limited to the convergence rate alone.
> While existing research, such as Zhang et al., only demonstrated convergence for the time-averaged policy $\frac1T\sum_{t=1}^T\pi^t$, we have proven last-iterate convergence,  which means the convergence of the actual policy $\pi^t$.
> Although we emphasize the novelty of our convergence rate, we do not claim novelty for the mirror descent method itself, as acknowledged in l.264, where we cite [Zhang, et al. 2023].
>
> One of the specific challenges that prior research, including [Zhang et al. 2023], did not address is the non-Lipschitzness of the Q function.
> We have mentioned this difficulty in our manuscript, particularly in l.340 and l.347.
> To overcome this challenge and improve the convergence result, we established a novel claim in Claim 4.5 and l.370: “the value function $ Q_h^{\lambda,\sigma} $ behaves well, almost as if it were a Lipschitz continuous function.”
>
> >* Additionally, providing numerical demonstrations of such exponential convergence is essential, given the magnitude of this improvement.
>
> Regarding the numerical demonstration of exponential convergence, we agree that including such figures would be beneficial.
> We plan to add these illustrations to the paper during the discussion period.
>
> >* Presentation: the presentation of the paper is not yet suitable for scientific publication. Please avoid excessive use of colors, shapes, and unconventional layouts. Figure 1 lacks informative content. Additionally, some language and structural transitions are too casual and do not provide sufficient discussion to support the arguments presented convincingly.
>
> Our intention was to enhance the visual clarity and engagement of the content. However, we understand the importance of maintaining a professional and conventional layout for scientific publications. We will revise the figures and layouts to ensure they adhere to presentation guidelines of ICLR.
>
> >* The numerical experiments are limited. More results regarding the algorithm performance in a comparison to baselines (OMD, FP, Multi-scale, double loop, single loop) for MFGs should be provided.
>
> The reason for conducting experiments on this example is to experimentally verify whether our proposed algorithm (Algorithm 2) converges in a typical MFG, not to compare it with existing methods.

---

> > ### Comment · Reviewer_AgLM · 2024-11-20
> >
> > Thanks for authors' clarifications. As I have mentioned in my previous comments, authors should conduct a comprehensive literature review to highlight their contributions or significance. Also, the experiments (scenarios, baselines) are not thorough to valid the proposed method. The current numerical results only serve as a verification. Therefore, I will keep my original score at this stage.

---

> ### Comment · Reviewer_AgLM · 2024-11-20
>
> Thanks for authors' responses. I am not convinced by the significance of the last-iterate convergence proposed in this work. Please check following works and highlight the research gap and contributions. A comprehensive review (including Xie and 2021 Zhang 2023) needs to be conducted by authors.
>
> 1. Mao 2022. A mean-field game approach to cloud resource management with function approximation
> 2. Zaman 2023. Oracle-free reinforcement learning in mean-field games along a single sample path
> 3. Yardim 2023. Policy mirror ascent for efficient and independent learning in mean field games
> 4. Zeng 2024. A Single-Loop Finite-Time convergent policy optimization algorithm for mean field games
> 5. Huang 2024. On the statistical efficiency of mean-field reinforcement learning with general function approximation.
> 6. Angiuli 2022. Unified reinforcement Q-learning for mean field game and control problems
> 7. Angiuli 2023. Convergence of multi- scale reinforcement q-learning algorithms for mean field game and control problems
>
> I will keep my original score at this stage, but I am open to discuss the significance of this work in a comparison to the missing papers I listed above.

---

> ### Author Response · Authors · 2024-11-21
>
> Thank you for your detailed feedback and for highlighting the additional references. We appreciate the opportunity to clarify the significance of our work in comparison to the mentioned papers. Below is a table summarizing the convergence results of the referenced works and our contributions:
>
> |                 | Learning algorithm               | Summary of convergence results                                               |
> |------------------------|----------------------------------|------------------------------------------------------------------------------|
> | Xie et al. (2021)      | Fictious play                    | time-averaging convergence                                                   |
> | Zhang et al. (2023)    | RMD                              | time-averaging convergence (to regularized equilibrium) under monotonicity   |
> | Mao et al. (2022)      | Actor-critic                     | time-averaging convergence (to regularized equilibrium)                      |
> | Zeman et al. (2023)    | Q-learning                       | time-averaging convergence (to regularized equilibrium)                      |
> | Yardim et al. (2023)   | Mirror Descent                   | LIC under contraction                                                        |
> | Zeng et al. (2024)     | Actor-critic                     | best-iterate convergence under Herding                                       |
> | Huang et al. (2024)    | Maximum Likelihood Estimation    | N/A                                                                          |
> | Angiuli et al. (2022)  | (two-time scale) Q-Learning      | N/A                                                                          |
> | Angiuli et al. (2024)  | (three-time scale) Q-learning    | LIC under contraction                                                        |
> | Pérolat et al. (2021)  | RMD                              | LIC under strict monotonicity in continuous time|
> | **Our work (Theorem 3.1)** | PP                           |  LIC under monotonicity                        |
> | **Our work (Theorem 4.4)** | RMD                          | LIC (to regularized equilibrium) under monotonicity                                                        |
>
> We have added this table to the revision as Table 1 in Appendix A.
>
> Based on this comparison, we would like to emphasize the following points:
>
> **Significance of Our Convergence Results**: Unlike many of the referenced works that require strong assumptions such as contraction to achieve convergence, our results demonstrate last-iterate convergence (LIC) without such stringent conditions. This highlights that our contributions fill a significant gap in the literature.
>
> We hope this clarifies the significance of our contributions and addresses the concerns raised. We are open to further discussion and appreciate your constructive feedback.

---

> > ### Comment · Reviewer_AgLM · 2024-11-22
> >
> > Thanks for the timely responses. Last-iterate convergence is a widely studied topic in optimization, and regularization is a widely studied topic in learning MFGs. Simply combining the two does not justify the significance of the work.

---

> > > ### Author Response · Authors · 2024-11-24
> > >
> > > Thank you for your timely responses and for recognizing the importance of last-iterate convergence (LIC) and regularized MFGs in our work.
> > > However, we respectfully disagree with the assertion that our work merely combines these two well-studied topics without significant contributions.
> > > Our work goes beyond a simple combination in the following ways:
> > >
> > > - **LIC of Proximal Point**: Demonstrating LIC for the proximal point method (PP) cannot be achieved by merely leveraging existing optimization techniques such as [[Censor & Zenios, 1992]](https://link.springer.com/article/10.1007/BF00940051).
> > >
> > > - **Our finding that the PP update is equivalent to RMD**: Moreover, our finding that a single update of PP can be interpreted as solving a crucial regularized MFG is non-trivial. Please refer to lines 180-185 of our manuscript for a detailed explanation.
> > >
> > > - **LIC of RMD**: Achieving exponential convergence rates to the regularized equilibrium is challenging with existing techniques. Our technical contributions include deriving discretization errors in Equation (4.6) that are distinct from those in policy optimization [[Zhan et al., 2023]](https://epubs.siam.org/doi/abs/10.1137/21M1456789) and regularized MFG [Zhang et al., 2023]. For a more detailed discussion, please refer to our rebuttal to Reviewer b31R.
> > >
> > > We hope this clarifies the novelty and significance of our contributions.

---

> > > > ### Comment · Reviewer_AgLM · 2024-11-26
> > > >
> > > > Thanks for the clarification. I have updated my score. Please include all these modifications or clarifications in the final manuscript and improve the presentation.

---

> > > > > ### Author Response · Authors · 2024-11-28
> > > > >
> > > > > Thank you for updating your rating. The comments you gave have already been reflected in the manuscript; we would be very grateful if you could also take a look at the global comments and reconsider your rating.

---

### Official Review · Reviewer_4hfX · 2024-11-04

**Soundness:** 3
**Presentation:** 2
**Contribution:** 2
**Rating:** 5
**Confidence:** 4

**Summary:**

This paper proposes and algorithm for mean field games (MFGs) based on proximal-point-type iterations. After proving convergence of the last iterate for an ideal algorithm, the paper considers an approximate algorithm based on mirror descent updates. For regularized MFGs, a proof of convergence is provided, with a rate of convergence. A simple illustrative example is shown at the end.

**Strengths:**

The paper is relatively clear and addresses an interesting question (last-iterate convergence). It seems to have a good theoretical foundation.

**Weaknesses:**

The first result holds under general assumptions but the algorithm cannot be implemented. The second result holds only for regularized MFGs, which can be quite different from the original (unregularized) MFG problem and there is not much discussion about this. As a consequence, the contributions seem relatively limited.

**Questions:**

Q1. In Assumption 2.1, please clarify why “It is reasonable to assume” it. For example, it is violated in many examples of the MFG literature in grid world models, where the agents cannot jump from a given state to any other given state (for any action). I think this would be true only if there is a very strong noise that can propel an agent to any other state in just one time step. It does not seem very realistic from the physical viewpoint. Please explain.

Q2. In Definition 4.2, are regularized equilibria unique? Otherwise, please explain how you can say “the regularized equilibrium” in Theorems 4.3 and 4.4.

Q3. Algorithm 2 and Theorem 4.3: it seems to rely very strongly on the RMD of Zhang et al. (2023). How does the proof of convergence of Theorem 4.3 differs from theirs?

Q4. Theorems 4.3 and 4.4: Could you please confirm whether these results hold for any $\lambda$ and $\sigma$, simply under Assumption 4.1? I just want to make sure there is no requirement to take $\lambda$ large enough as e.g., in  Cui & Koeppl (2021).

Q5. Theorems 4.3 and 4.4: Here it seems that the convergence only holds for the regularized MFG. How far is it from the true MFG equilibrium?

Q6. Could you please explain whether the example considered in the numerical part satisfies all the assumptions required for your convergence results? Also, is it not strictly monotone (so that it does not fall in the case covered by Perolat et al. (2022))?

Q7. To really see how your method compares with existing one, please compare your algorithm with other baselines, such as the algorithms considered in Perolat et al. (2022) (in particular the mirror-descent-type one, which has last-iterate convergence, as you mentioned).

Typos:
Definition 2.4: “is mean-field” -> “is a mean-field”
Page 3: “has the full support” -> “has full support”

---

> ### Author Response · Authors · 2024-11-20
> **Rebuttal by authors (Part 1)**
>
> Thank you for your review.
> We will address your comments below:
>
> >* Weaknesses: The first result holds under general assumptions but the algorithm cannot be implemented. The second result holds only for regularized MFGs, which can be quite different from the original (unregularized) MFG problem and there is not much discussion about this. As a consequence, the contributions seem relatively limited.
>
> First of all, we would like to emphasize again that we use the regularized mirror descent of the second result to approximate the proximal point method in the first result for solving (original) MFG.
> Moreover, __this paper is the first work to provide the last-iterate convergence guarantee for solving original MFGs under the monotonicity condition__.
>
> We acknowledge that there are differences between regularized MFGs and unregularized MFGs.
> In fact, in Algorithm 2, we use regularized MFGs merely as a subroutine to solve the original unregularized MFG problem.
> More importantly, our contribution includes the proposal of an algorithm that integrates mirror descent for regularized MFGs into a proximal point method for unregularized MFGs, whose convergence has been experimentally verified. This result extends our theoretical findings into a practical algorithm, demonstrating the applicability and effectiveness of our approach.
>
> >* Questions: Q1. In Assumption 2.1, please clarify why “It is reasonable to assume” it. For example, it is violated in many examples of the MFG literature in grid world models, where the agents cannot jump from a given state to any other given state (for any action). I think this would be true only if there is a very strong noise that can propel an agent to any other state in just one time step. It does not seem very realistic from the physical viewpoint. Please explain.
>
> First, we guess that even the grid world model example you mentioned satisfies Assumption 2.1.
> In this case, we can exclude the states from which no transitions are possible from the state set $\mathcal{S}$.
> By doing so, the remaining states will satisfy the assumption.
>
> Assumption 2.1 states that "for any state $ s^\prime $, __there exists__ a state-action pair $ (s, a) $ such that $ s^\prime $ is reachable."
> It does not require that "for any state $ s^\prime $, it is reachable by __any__ state-action pair $ (s, a) $". This distinction is crucial, because it only requires the existence of at least one state-action pair that can reach a given state, not that all state-action pairs can reach the state.
>
> >* Q2. In Definition 4.2, are regularized equilibria unique? Otherwise, please explain how you can say “the regularized equilibrium” in Theorems 4.3 and 4.4.
>
> Yes, the regularized equilibria are unique.
> This uniqueness is discussed in l.245 and Proposition B.1 of our manuscript.
>
> To elaborate further: Suppose there are two different regularized equilibria $(\mu^\ast_1, \varpi^\ast_1)$ and $(\mu^\ast_2, \varpi^\ast_2)$. If we assume $\varpi^\ast_1 \neq \varpi^\ast_2$, the following contradiction arises:
>     - From Lemma A.2, which uses monotonicity, we have $$ J^{\lambda,\sigma}(\mu^\ast_1, \varpi^\ast_1) + J^{\lambda,\sigma}(\mu^\ast_2, \varpi^\ast_2) \leq J^{\lambda,\sigma}(\mu^\ast_1, \varpi^\ast_2) + J^{\lambda,\sigma}(\mu^\ast_2, \varpi^\ast_1) .$$
>     - Additionally, from Proposition B.1, we know that $ J^{\lambda,\sigma}(\mu^\ast_1, \varpi^\ast_1) \gneq J^{\lambda,\sigma}(\mu^\ast_1, \varpi^\ast_2) $ and $ J^{\lambda,\sigma}(\mu^\ast_2, \varpi^\ast_2) \gneq J^{\lambda,\sigma}(\mu^\ast_2, \varpi^\ast_1) $. Adding these two inequalities gives us $$ J^{\lambda,\sigma}(\mu^\ast_1, \varpi^\ast_1) + J^{\lambda,\sigma}(\mu^\ast_2, \varpi^\ast_2) \gneq J^{\lambda,\sigma}(\mu^\ast_1, \varpi^\ast_2) + J^{\lambda,\sigma}(\mu^\ast_2, \varpi^\ast_1) .$$
>
> Therefore, $\varpi^\ast_1 = \varpi^\ast_2$.
> Moreover, by the definition of regularized equilibria, $\mu^\ast_1 = m[\varpi^\ast_1] = m[\varpi^\ast_2] = \mu^\ast_2$.
> This contradicts the assumption that the two equilibria are different.
> Thus, the equilibrium is unique.

---

> ### Author Response · Authors · 2024-11-20
> **Rebuttal by authors (Part 2)**
>
> >* Q3. Algorithm 2 and Theorem 4.3: it seems to rely very strongly on the RMD of Zhang et al. (2023). How does the proof of convergence of Theorem 4.3 differs from theirs?
>
> Before clarifying how Zhang et al.'s arguments in their proof differ from ours, let us also emphasize how the __results__ differ.
>
> In contrast to us, Zhang et al. (2023)'s result does not imply last-iterate convergence.
> It would also be difficult to obtain a result of last-iterate convergence with rate $\mathcal{O}(T^{-1/2})$ using Zhang et al. (2023)'s argument.
>
> In order to overcome this difficulty and achieve exponential convergence, we came up with a completely different strategy to that used in the proof of  [Theorem 5.4, Zhang et al. 2023].
> The key to the proof of Theorem 4.3 lies in the claim stated in l.369-372: “over the sequence $(\pi^t)_t$, the value function $ Q_h^{\lambda,\sigma} $ behaves well, almost as if it were a Lipschitz continuous function.”
> In contrast, Zhang et al. do not utilize this Lipschitz continuity.
> Instead, they bound the difference in $ Q $ by a (potentially very large) constant; see [Eq. (E.1), Zhang et al.].
> By leveraging the Lipschitz-like behavior of $ Q_h^{\lambda,\sigma} $ over the sequence of policies, our approach provides a more refined analysis, leading to the convergence results stated in Theorem 4.3.
>
> >* Q4. Theorems 4.3 and 4.4: Could you please confirm whether these results hold for any λ and σ, simply under Assumption 4.1? I just want to make sure there is no requirement to take λ large enough as e.g., in Cui & Koeppl (2021).
>
> Yes, Theorems 4.3 and 4.4 hold simply under Assumption 4.1.
> Additionally, there is no need to take $\lambda$ to be large enough, as required in Cui & Koeppl (2021).
> We would like to note that Cui & Coeppl adopt different assumptions from ours. For example, they do not impose the monotonicity of reward.
> >* Q5. Theorems 4.3 and 4.4: Here it seems that the convergence only holds for the regularized MFG. How far is it from the true MFG equilibrium?
>
> Roughly speaking, the distance between the two equilibria is $\mathcal{O}(\lambda)$.
> More precisely, for the regularized equilibrium $(\mu^\ast, \varpi^\ast)$ and the original equilibrium $(\mu^\star, \pi^\star)$, we have $ J(\mu^\star, \pi^\star) - J(\mu^\ast, \varpi^\ast) = \mathcal{O}(\lambda) $.
>
> >* Q6. Could you please explain whether the example considered in the numerical part satisfies all the assumptions required for your convergence results? Also, is it not strictly monotone (so that it does not fall in the case covered by Perolat et al. (2022))?
> >* Q7. To really see how your method compares with existing one, please compare your algorithm with other baselines, such as the algorithms considered in Perolat et al. (2022) (in particular the mirror-descent-type one, which has last-iterate convergence, as you mentioned).
>
> The example used in our experiments satisfies all the assumptions required for our convergence results.
> Specifically, it is a monotone example.
> The reason we conducted experiments on this example was to experimentally verify whether Algorithm 2, which is a novel combination of proximal point with tractable approximation, converges in a typical MFG setting rather than to compare it with existing methods.

---

> > ### Author Response · Authors · 2024-11-28
> >
> > Thank you for taking the time to review our paper.
> > We have revised the manuscript to strengthen the survey of previous studies so that the novelty of our paper becomes clear: please also see the global comments. We would like you to check these comments and reconsider your rating.

---

> > > ### Comment · Reviewer_4hfX · 2024-12-01
> > > **Response**
> > >
> > > Thank you for your detailed reply and the efforts to improve the paper.

---

### Official Review · Reviewer_YwpQ · 2024-11-21

**Soundness:** 3
**Presentation:** 3
**Contribution:** 2
**Rating:** 5
**Confidence:** 4

**Summary:**

This paper proposes algorithms for achieving fast convergence to equilibria in monotone mean field games. The first (conceptual) algorithm is based on a proximal point method type of algorithm that is based on possibly hard to implement update step. Next, the authors employ the Mirror Descent algorithm for a regularized MFG which can efficiently approximate the update of the proximal point method. This process converges exponentially fast to equilibria of the regularized game.

**Strengths:**

- the paper studies a well motivated problem (MFG games + monotonicity).
- it contains provable theoretical analysis of their last iterate results.
- the proof structure is easy to parse.

**Weaknesses:**

- the theoretical results feel somewhat incremental from a technical stand point. E.g. the use of the regularized variant of MD to achieve fast convergence in monotone settings feel very similar to e.g. [1]. Clearly, there are more technicalities involved to adapt the results to this precise setting but arguably these technicalities do not feel conceptually/technically novel. On the somewhat positive side, this makes parsing the proof rather straightforward.
- the experiments could be expanded. For example, prior work that studied last-iterate convergence in (non-strict) monotone settings used the experiments to study the effects of equilibrium selection [2]. Although the possibility of diverse equilibria is advertised in the paper as an important element of monotone games, not present in strictly monotone games, the implications of which equilibria get selected is not addressed. I feel that this is a missed opportunity that would make the paper significantly stronger.

[1] Julien Perolat, et al. From Poincare recurrence to convergence in imperfect information games: Finding equilibrium via regularization. In ICML, volume 139 of Proceedings of Machine Learning Research, pp. 8525–8535. PMLR, 2021.
[2] Leonardos, S., et al. Exploration-exploitation in multi-agent competition: convergence with bounded rationality. Advances in Neural Information Processing Systems, 2021.

**Questions:**

Can you say anything about the effects of your algorithm to equilibrium selection? Either theoretically or experimentally?

---

> ### Author Response · Authors · 2024-11-22
> **Rebuttal by authors**
>
> Thank you for your additional feedback. We would like to address your points in detail:
>
> **Regarding the theoretical results feeling somewhat incremental:**
> >*  the theoretical results feel somewhat incremental from a technical stand point. E.g. the use of the regularized variant of MD to achieve fast convergence in monotone settings feel very similar to e.g. [1]. Clearly, there are more technicalities involved to adapt the results to this precise setting but arguably these technicalities do not feel conceptually/technically novel. On the somewhat positive side, this makes parsing the proof rather straightforward.
>
> We respectfully disagree with the assertion that our theoretical results are incremental.
> The work you referenced, [1] by Perolat et al., discusses the convergence of a continuous-time algorithm (FoRL) for Sequential Imperfect Information Games.
> Our setting and their setting differ significantly in the following ways:
> - **Sequential Imperfect Information Game vs. Mean Field Game (MFG):** While both frameworks define a probability distribution relative to a policy $\pi$, Perolat et al. focus on the reach probability $\rho^\pi$ over __histories__, whereas we focus on the distribution of __states__ $\mu=m[\pi]$. The dependency on $\pi$ is fundamentally different: $\rho$ depends on $\pi$ in a linear-like manner, while our $\mu$ has a highly nonlinear dependency on $\pi$ as shown in equation (2.1). Addressing this nonlinearity necessitated the development of novel techniques that exploit the inductive structure of equation (2.1) with respect to time $h$.
> - **Continuous-time vs. Discrete-time Analysis:** Perolat et al. primarily analyze the convergence of the continuous-time FoRL algorithm. In contrast, our work extends beyond continuous-time convergence (Theorem 4.4) to also address discrete-time convergence (Theorem 4.3). Generally, proving convergence in discrete-time is more challenging than in continuous-time. For example, the proof of Theorem 4.3 requires precise evaluation of discretization errors, which do not arise in continuous-time analysis; see equation (4.6) and Claim 4.5 for details.
>
> **Regarding equilibrium selection:**
>
> >* the experiments could be expanded. For example, prior work that studied last-iterate convergence in (non-strict) monotone settings used the experiments to study the effects of equilibrium selection [2]. Although the possibility of diverse equilibria is advertised in the paper as an important element of monotone games, not present in strictly monotone games, the implications of which equilibria get selected is not addressed. I feel that this is a missed opportunity that would make the paper significantly stronger.
> >* Questions: Can you say anything about the effects of your algorithm to equilibrium selection? Either theoretically or experimentally?
>
> Within the scope of our Theorem 3.1, we can guarantee convergence to the set of equilibria, where the distance to this set converges to zero. Experimentally, we can observe that our algorithm indeed selects specific equilibria, converging to one of them. We hypothesize that the equilibrium closest to the initial policy $\pi^0$ in terms of KL divergence is selected. This hypothesis and its implications for equilibrium selection represent a promising direction for future research.

---

> > ### Author Response · Authors · 2024-11-28
> >
> > Thank you for taking the time to review this manuscript. We have revised the manuscript in response to your comments (see global comments). We would be grateful if you could check these comments and reconsider your rating.

---

> > > ### Comment · Reviewer_YwpQ · 2024-11-28
> > > **Response**
> > >
> > > I thank the authors for their response. After reading it as well as the comments from the other reviewers I maintain my score.

---

### Author Response · Authors · 2024-11-28
**Global  comment by authors**

We appreciate the detailed feedback on our paper. We would like to highlight the technical novelty of our main theorems (Theorem 3.1 and 4.3).

---

# Significance of Our Convergence Results
First, we would like to emphasize again the importance of our convergence results.

- **LIC without Strong Assumptions**: Unlike many referenced works requiring strong assumptions (e.g., contraction), our results demonstrate last-iterate convergence (LIC) without such conditions.
- **Proximal Point Method (PP)**: Demonstrating LIC for PP cannot be achieved by merely leveraging existing optimization techniques. Our finding that a single update of PP can be interpreted as solving a crucial regularized MFG is non-trivial (see lines 180-185 of our manuscript).
- **Exponential Convergence of Regularized Mirror Descent**: Derived a different discretization error $ D_{\mu^\ast}(\pi^t, \pi^{t+1}) $ from Zhang et al., facilitating clearer subsequent analysis. Our discretization error can be bounded by $ D_{\mu^\ast}(\pi^\ast, \pi^t) $ (see l.369-372).

---


# Comparison with Prior Work
In addition to Appendix A, we discuss the comparison with specific previous studies as indicated by the reviewers below:
- **Comparison with Zhang et al. (2023)**
  - **Zhang et al.**: Demonstrated convergence for the time-averaged policy $\frac{1}{T}\sum_{t=1}^T\pi^t$ with the rate $\mathcal{O}(T^{-1/2})$.
  - **Our Work**: Proven last-iterate convergence with the exponential rate, meaning the convergence of the actual policy $\pi^t$. This provides stronger guarantees on the convergence behavior of the algorithm.

- **Continuous-time Analysis in Perolat et al. (2022) vs. Discrete-time Analysis**
  - **Perolat et al.**: Analyzed the convergence of the continuous-time Follow-the-Regularized-Leader algorithm.
  - **Our Work**: Extended beyond continuous-time convergence (Theorem 4.4) to address discrete-time convergence (Theorem 4.3). Discrete-time convergence is more challenging due to the need for precise evaluation of discretization errors (see equation (4.6) and Claim 4.5 for details).

- **Sequential Imperfect Information Game in Perolat et al. (2021) vs. Mean Field Game (MFG)**
  - **Perolat et al.**: Focused on the reaching probability $\rho^\pi$ over histories in Sequential Imperfect Information Games.
  - **Our Work**: Focused on the distribution of states $\mu=m[\pi]$ in MFGs. The dependency on $\pi$ is fundamentally different: $\rho$ depends on $\pi$ in a linear-like manner, while our $\mu$ has a highly nonlinear dependency on $\pi$ (see equation (2.1)). Addressing this nonlinearity required novel techniques exploiting the inductive structure of equation (2.1) with respect to time $h$.
  > Julien Perolat, et al. From Poincare recurrence to convergence in imperfect information games: Finding equilibrium via regularization. In ICML, volume 139 of Proceedings of Machine Learning Research, pp. 8525–8535. PMLR, 2021.

- **Markov Decision Processes (MDPs) in Zhan et al. (2023) vs. MFGs**
  - **Zhan et al.**: The theory for MDPs cannot be directly applied to MFG for the convergence of mirror descent. The inner product $\langle Q^{k}(s),\pi^{k+1}(s)-p\rangle$ in [[Lem. 6, Zhan et al.]](https://arxiv.org/abs/2105.11066) is not compatible with the property specific to MFG (lemma E.4).
  - **Our Work**: Derived $\langle Q^{k}(s),\pi^{k}(s)-p \rangle$ instead, allowing us to apply the property (see the fourth line of equation (D.6)).
  > Zhan, W., Cen, S., Huang, B., Chen, Y., Lee, J. D., & Chi, Y. (2023). Policy mirror descent for regularized reinforcement learning: A generalized framework with linear convergence. SIAM Journal on Optimization, 33(2), 1061-1091.

---


# Our Technical Novelty
Finally, we explain the techniques of proof that we used to obtain the novel results described above.

- **Challenge**: Non-Lipschitzness of the Q function, not addressed by prior research including Zhang et al. (2023).
- **Our Solution**: Established a novel claim in Claim 4.5 and l.370: “the value function $ Q_h^{\lambda,\sigma} $ behaves well, almost as if it were a Lipschitz continuous function.”

---

We hope this clarifies the technical novelty and significance of our contributions.

---

> ### Author Response · Authors · 2024-11-28
> **Revised parts of the manuscript**
>
> We have revised the manuscript to make the above points more clear. The revised parts are highlighted in purple. The main revisions are as follows.
> - In Appendix A, we have listed a wider range of related works. Specifically, we have also compared our paper with previous research on MFG algorithms that are not limited to mirror descent.
> - In Figure 2, we have experimentally confirmed that Regularized Mirror Descent converges exponentially.

---

### Meta-Review · Area_Chair_xyDr · 2024-12-18

**Metareview:**

This paper examines a proximal-point algorithm for computing equilibrium strategies in mean field games (MFGs), and focuses in particular on the algorithm's "last-iterate" convergence properties - that is, the convergence of the sequence of play generated by the algorithm, as opposed to some averaged version thereof. The authors consider MFGs that satisfy a (weak) monotonicity condition in the spirit of Lasry and Lions, and they employ an approach based on mirror descent to efficiently approximate update rules for regularized MFGs. The paper's main result is that regularized mirror descent converges at a geometric rate, attaining an $\varepsilon$-equilibrium in $\mathcal{O}(\log(1/\varepsilon))$ iterations.

While the reviewers appreciated the convergence properties of the proposed algorithms, it was pointed out that the combination of regularization and Bregman-type proximal point algorithms is very close to previous works in the literature - e.g., by Perolat et al. in the context of MFGs - and the effort required to adapt these techniques to the authors' precise setting was deemed somewhat incremental. Concerns were also raised regarding the experimental validation of the proposed algorithms relative to other algorithmic baselines for MFGs, especially in the strict-/weak-monotonicity gap.

The authors' rebuttal and responses did not significantly sway the reviewers' original opinion, who remained unanimously on the side of rejection (albeit borderline so), ultimately leading to a reject recommendation.

**Additional Comments On Reviewer Discussion:**

To conform to ICLR policy, I am repeating here the relevant part of the metareview considering the reviewer discussion:

> While the reviewers appreciated the convergence properties of the proposed algorithms, it was pointed out that the combination of regularization and Bregman-type proximal point algorithms is very close to previous works in the literature - e.g., by Perolat et al. in the context of MFGs - and the effort required to adapt these techniques to the authors' precise setting was deemed somewhat incremental. Concerns were also raised regarding the experimental validation of the proposed algorithms relative to other algorithmic baselines for MFGs, especially in the strict-/weak-monotonicity gap.
>
> The authors' rebuttal and responses did not significantly sway the reviewers' original opinion, who remained unanimously on the side of rejection (albeit borderline so).

---

### Decision · Program_Chairs · 2025-01-22

Reject